# Connectome-Based Modelling Reveals Orientation Maps in the *Drosophila* Optic Lobe

**Jia-Nuo Liew**[1,2,*]**, Shenghan Lin**[3,*]**, Bowen Chen**[2,4,6,7]**, Wei Zhang**[1,2]**,**
**Xiaowei Zhu**[4]**, Wei Zhang**[2,4,6,7,†]**, Xiaolin Hu**[1,2,8,†]

[1]Department of Computer Science and Technology, BNRist, Tsinghua University, Beijing 100084, China
[2]IDG/McGovern Institute of Brain Research, Tsinghua University, Beijing 100084, China
[3]Zhili College, Tsinghua University, Beijing 100084, China
[4]School of Life Sciences, Tsinghua University, Beijing 100084, China
[5]Ant Group, China
[6]State Key Laboratory of Membrane Biology, Tsinghua University, Beijing 100084, China
[7]Tsinghua-Peking Center for Life Sciences, Tsinghua University, Beijing 100084, China
[8]Chinese Institute for Brain Research (CIBR), Beijing 100010, China

```
{liujn24, linsh24, cbw21, zhangw23}@mails.tsinghua.edu.cn
                 robert.zxw@antgroup.com
       {wei_zhang, xlhu}@mail.tsinghua.edu.cn
```

## Abstract

The ability to extract oriented edges from visual input is a core computation across animal vision systems. Orientation maps, long associated with the layered architecture of the mammalian visual cortex, systematically organise neurons by their preferred edge orientation. Despite lacking cortical structures, the *Drosophila melanogaster* brain contains feature-selective neurons and exhibits complex visual detection capacity, raising the question of whether map-like vision representations can emerge without cortical infrastructure. We integrate a complete fruit fly brain connectome with biologically grounded spiking neuron models to simulate neuroprocessing in the fly visual system. By driving the network with oriented stimuli and analysing downstream responses, we show that coherent orientation maps can emerge from purely connectome-constrained dynamics. These results suggest that species of independent origin could evolve similar visual structures.

## 1 Introduction

The ability to extract oriented edges from visual input is a core computation across animal vision systems [23]. Orientation selectivity is canonically attributed to the primary visual cortex (V1), where neurons respond selectively to specific edge orientations. Hubel and Wiesel [22, 23] first demonstrated this modular structure in cats. The findings were later substantiated by electrophysiology, optical imaging, and detailed anatomical mapping [3, 41]. These orientation-selective responses are embedded within columnar and laminar architecture and are thought to arise from a combination of spatially organised feedforward inputs and recurrent cortical dynamics [2, 12, 15, 19, 48].

Orientation selectivity has been directly observed in *Drosophila melanogaster* visual system. In particular, T4 and T5 neurons - traditionally known for direction selectivity - also exhibit robust orientation tuning, which sharpens motion detection [16]. These findings provide physiological

---

[*]Equal contribution.
[†]Corresponding author.

39th Conference on Neural Information Processing Systems (NeurIPS 2025).

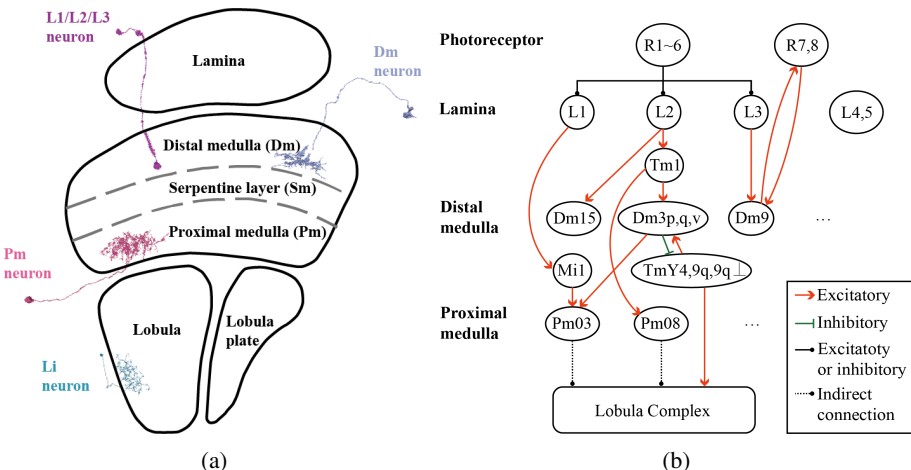

Figure 1: **Anatomical organisation and connectivity in the fly visual system.** (a) Schematic of the fly optic lobe showing the layered structure of the medulla, including the distal (Dm), serpentine (Sm), and proximal (Pm) sublayers. (b) Simplified connectivity diagram highlighting the visual pathway from photoreceptors (R1-6) through lamina neurons (L1-L3), to the orientation-selective neurons, to the lobula complex.

evidence that orientation selectivity exists at the neuron level. Building on this, recent studies have hypothesised that neurons in early visual stages, particularly in the medulla, could support orientation selectivity, based solely on synaptic connectivity patterns from structural data [45]. Though the question of whether such orientation selectivity is organised into coherent maps across the visual system remains unknown. Given the compact, non-layered architecture of the *Drosophila* brain and the absence of large-scale recurrent loops seen in vertebrate cortex, it is unclear whether this system can support the emergence of global tuning structures such as orientation maps [7, 10, 34].

To investigate this, we simulated the visual responses of early-stage neurons (L1-L3) using bar-like stimuli and propagated their activity through a downstream population in the optic lobe modelled with leaky integrate-and-fire (LIF) dynamics, constrained by known synaptic connections [13]. We then analysed the resulting neural population activity, revealing spatially organised orientation tuning reminiscent of cortical orientation maps. Figure 1a provides an overview of the anatomical structure of the *Drosophila* optic lobe, while Figure 1b illustrates the visual connectivity pathway examined in this study.

To summarise, our main contributions are as follows:

- We computationally demonstrated, for the first time, spatially coherent orientation maps in the medulla region of an invertebrate visual system.
- We identified topological singularities and inter-layer columnar alignment in orientation preference within the distal medulla (Dm) and proximal medulla (Pm) regions.

Together, these findings suggest that canonical orientation maps can arise from shared computational principles across species, even in the absence of cortical lamination.

## 2 Background and related works

**Orientation maps in mammals.** Orientation selectivity has been extensively studied in the visual cortices of many mammals. At the anatomical level, evidence from mammals indicates that orientation tuning can emerge in neural substrates with vast differences [25, 4, 38]. In rodents, maps often display salt-and-pepper rather than columnar architecture, as observed in the visual cortex, while the retina displays a continuous topographic map of orientation tuning [48]. In contrast, cats and wallabies display a structured pinwheel-like feature that has been documented in the primary visual cortex [24, 25, 37], despite differences in cortical evolutionary divergence across mammalian species [41]. These findings suggest that orientation maps can arise across diverse mammalian species with varying

cortical architectures, raising the possibility that such maps are not an exclusive feature of the neocortex but reflect convergent computational motifs.

**Orientation tuning in non-mammals.** Studies across non-mammalian species suggest that structured visual coding arises under minimal anatomical constraints. In pigeons, sharply tuned orientation-selective neurons emerge through feedforward circuits alone, without a laminated cortex [32]. In turtles, despite the absence of fine retinotopy, population activity in the dorsal cortex accurately encodes spatial information [17], indicating that a global analysis of the visual scene can arise from distributed representations. Similarly, in zebrafish, orientation-selective neurons have been observed in the optic tectum, and some studies report their laminar segregation and retinotopic organisation [40, 43]. However, while all three species exhibit orientation-selective neurons, neither has demonstrated the presence of continuous, spatially organised orientation maps comparable to the ones found in the mammalian cortex.

**Orientation coding in Drosophila.** In *Drosophila melanogaster*, orientation tuning has been hypothesised based on anatomical structure alone. In particular, Seung [45] predicted that functionally distinct neuron types such as Dm3 and TmY may exhibit orientation selectivity, proposing that such responses could arise purely based on wiring alone. To date, the only direct physiological evidence of orientation selectivity comes from earlier studies on T4 and T5 neurons, which exhibit tuning to oriented edges in addition to their well-known direction selectivity [16]. Beyond these, no other neuron types have been experimentally or computationally confirmed to show orientation selectivity. Moreover, no prior study has computationally demonstrated the emergence of orientation maps in this non-cortical system. Our work fills in this gap by building on these findings to show that coherent, spatially organised orientation maps can emerge in such a system, thus suggesting that species of vastly different evolutionary origin may share common circuit-level principles for encoding visual features.

# 3    Methods

**LIF model.** We implemented a leaky integrate-and-fire (LIF) framework to simulate the spiking dynamics of neurons in the *Drosophila* visual system (see Appendix B.1) [46]. Using the full adult fly brain (FAFB) connectome, we constructed a network in which neurons are labelled and connected according to their anatomical synapses [8, 13, 20, 50]. The model encompasses the complete adult *Drosophila* connectome, including 138,639 neurons and 1,508,983 synapses. The FAFB connectome provides a complete and cell-resolved reconstruction of the *Drosophila* brain [50], and has become a foundational resource for structural annotation [44], functional inference [49, 11], and whole-brain spiking simulations validated against behaviour [46]. In our model, visual stimuli were simulated as Poisson spike trains injected into lamina inputs (L1-L3) in the right eye of the *Drosophila*, and the resulting activity was propagated through the whole brain connectome using the embedded LIF framework, allowing us to monitor orientation tuning in downstream visual neurons (Figure 2a).

**Stimuli.** We focused on the lamina neurons L1-L3, which are visual columnar neurons - retinotopically organised neurons that are associated with an individual ommatidium in the compound eye, such that each visual column contains a dedicated copy of the neuron [39]. Unlike other columnar neurons, L1-L3 receive direct input from photoreceptors R1-R6, which capture brightness change across the visual field. The ommatidia themselves are arranged in a hexagonal lattice, forming a precise sampling grid over visual space. For this study, we applied input directly to L1-L3 rather than R1-R6, as the connectomic dataset provides visual columnar mappings for L1-L3 but not for photoreceptors [8, 13, 14, 20, 33, 35, 36, 44, 50, 39]. Naively, one can let an L1-L3 neuron receive a fixed strength of Poisson input regardless of the location of the OFF-bar on the receptive field (RF) of the corresponding ommatidium. However, this ignored spatial gradients across the RF. To better reflect spatial structure, the input stimuli were modelled by computing the distance $d$ from each ommatidium to the bar and making the input strength depend on $d$ based on the calcium imaging data [29] (Appendix B.2). Figure 2b illustrates how the distance of the OFF-bar modulates input strength. In the top row, each panel shows a single ommatidium, colour-coded by its corresponding L1-L3 firing rate as a function of $d$ from the OFF-bar (black line). Firing rates increased as the bar approached the centre of the ommatidia, peaking when aligned and tapering off with distance. The

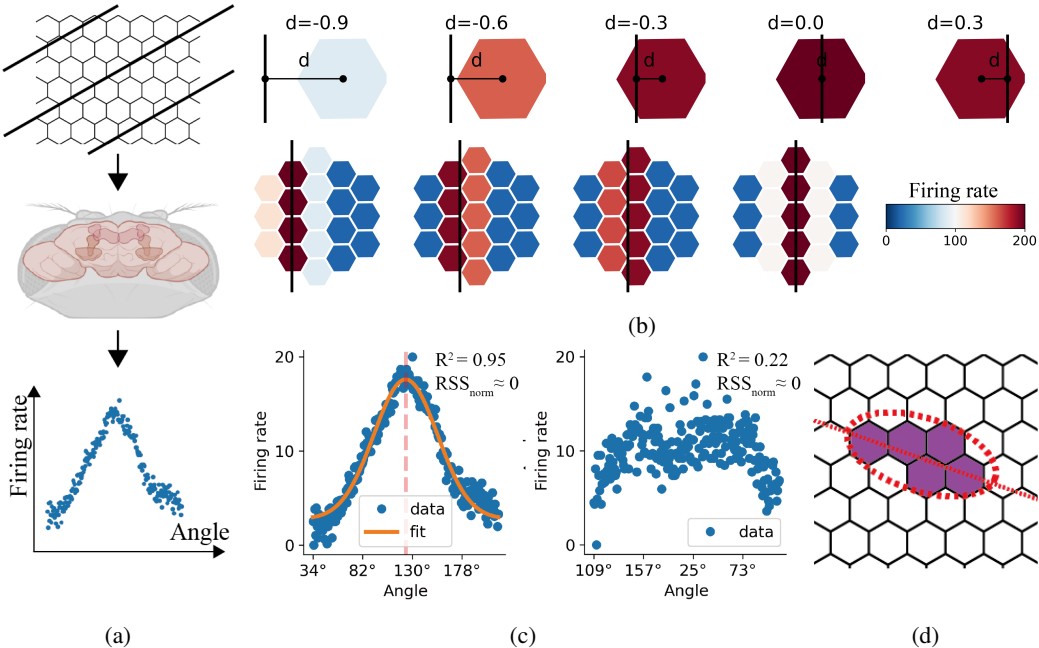

Figure 2: **Overview of the simulation framework and analysis methods.** (a) Schematic of the simulation pipeline. Poisson spike trains were used to simulate grating stimuli input to L1-L3 neurons. Each neuron's orientation tuning is quantified by its firing rate across stimulus angles. (b) Structured stimulus input to L1-L3 neurons across different OFF-bar positions (black line). Top: Each panel shows a hexagon representing an ommatidium. Bottom: 2D heatmap of firing activity across the simplified hexagonal lattice of ommatidia. Each hexagon represents one ommatidium, colour-coded by its corresponding L1-L3 firing rate. $d$ represents the distance to the centre of the ommatidia. Please note that, even when the bar stimulus lies outside the geometric boundary of an ommatidium, a measurable response is still observed due to the broader receptive field ( $5°$- $8°$) relative to the ommatidial spacing ($4.6°$), resulting in spatial overlap of sensitivity across neighbouring units[5]. (c) Gaussian turning curves of two neurons: the left panel indicates a well-fit neuron (orientation-selective), and the right panel indicates a neuron classified as not orientation-selective. (d) Structural prediction of preferred orientation angles based on upstream columnar connectivity. The red dashed lines indicate the best-fit ellipse drawn through these spatial locations.

bottom row provides a simplified illustration of the 2D activation pattern across the hexagonal lattice for several OFF-bar positions.

**Preferred orientation.** To quantify the orientation selectivity of the monitored neurons, we fit a Gaussian function to their firing responses across different stimulus orientations. Unlike a standard Gaussian, which assumes a linear domain, our model accounts for the $180°$ periodicity of angle space by computing the minimal circular distance between the stimulus orientation and the neuron's preferred orientation (Figure 2c; see Appendix B.3). A fit was considered "good" if it satisfied two criteria: a coefficient of determination $R^2 \geq 0.7$ and a normalised residual sum of squares $RSS_{norm} \leq 0.4$ – indicating both high explanatory power and low relative error (Figure 2c). Representative examples of Gaussian fits that are considered good, poor and near threshold are illustrated in Figure S4a. The distribution of $R^2$ across neurons and the relationship between $R^2$ and $RSS_{norm}$ are illustrated in Figure S4b, S4c. Unless otherwise noted, the preferred orientation of a neuron in this paper refers to its orientation preference as determined by this Gaussian fit.

**Structural prediction.** In this paper, structured prediction refers to the predicted preferred orientation angle of a neuron by its direct upstream neurons, similar to the receptive field approach used by Seung [45]. Specifically, we identify each neuron's upstream partners that are also columnar neurons using known synaptic connectivity. We mapped these upstream neurons to their corresponding columns in the compound eye, and a best-fit ellipse is drawn around their spatial locations (Figure 2d).

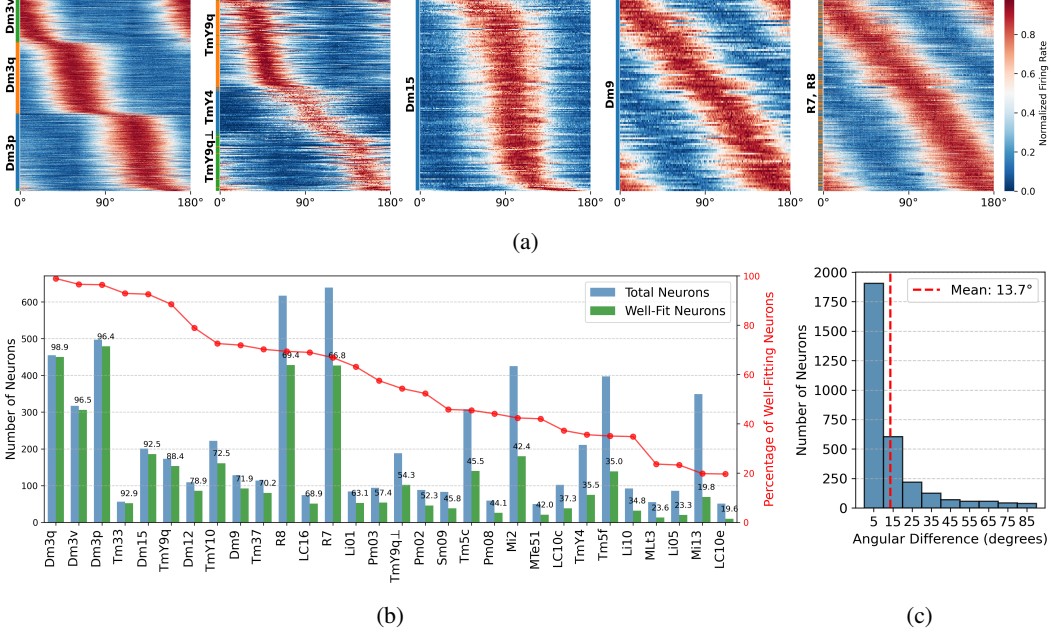

Figure 3: **Orientation tuning and structural prediction accuracy.** (a) Heatmaps showing the normalised firing rates of various neurons across different angular orientations. (b) Proportion of neurons with well-fit orientation tuning across types. Green bars indicate the subset well-fitted by a circular Gaussian model (see methods). Blue bars indicate the total number of neurons in the right optic lobe of the respective type. (c) Distribution of absolute differences between each neuron's preferred orientations predicted by LIF simulation and by dendritic structure.

The orientation of the ellipse's major axis, measured relative to the y-axis, is taken as the predicted angle for the downstream neuron.

## 4 Results

### 4.1 Orientation selectivity

We first evaluated whether neurons in the *Drosophila* optic lobe exhibit robust orientation selectivity. Using our LIF-based simulation framework, we presented bar-like stimuli across orientations and recorded their firing rates. We then fitted each neuron's response profile using a circular Gaussian model (see Methods). For neurons that were well-fit, we normalised firing rates to their peak responses and visualised the resulting heatmap of preferred orientations. Neurons were initially sorted via hierarchical clustering, revealing structured patterns in tuning preferences. Upon observing a diagonal-like structure - suggestive of a continuous orientation gradient - we further sorted neurons by the ascending position of their peak response, which highlighted a clearer gradient in orientation tuning (Figure 3a). Our findings demonstrated that *Drosophila* neurons exhibit robust orientation selectivity, with evidence suggesting that orientation maps in this species may exist. The percentage of well-fit neurons relative to the total neurons in the type for a subset of neurons (Dm3v, Dm3p, Dm3q, TmY4, TmY9q, TmY9q$^\perp$) aligns well with Seung's prediction [45], indicating that orientation selectivity arises directly from the spatial arrangement of dendritic inputs (left two panels of Figure 3a).

Moreover, we found that several other neuron types (e.g., Dm15, Tm33, TmY10) also exhibited tuning properties (Figure 3a). To quantify this selectivity, we classified a neuron type as orientation selective if over 40% of its neurons were well fit (Figure 3b; see Methods). These results revealed that most orientation-selective neurons were located within the medulla layer, highlighting it as a key site for early orientation processing in the *Drosophila* visual system. To further investigate the tuning mechanism and verify our hypothesis that orientation tuning could be directly influenced

by its dendritic inputs, we quantified the preferred orientation of each neuron by computing the structural prediction of the neuron (Figure 2a; see Methods). We then compared it to the neuron's preferred orientation using absolute difference. Across the population analysed, the mean absolute difference sits at $13.7°$ (Figure 3c), indicating a robust correspondence between dendritic geometry and orientation preference. Notably, T4 and T5 neurons, despite being known for their direction and orientation selectivity, did not meet our selection criteria in this analysis. Our results showed that most T4 and T5 neurons had firing rates close to zero. A few neurons that had higher firing rates exhibited orientation selectivity (Figure S5). This was likely due to the stimulus design: the static input may have been insufficient to strongly activate these neurons.

Intriguingly, while analysing the orientation tuning patterns across neuron populations, we incidentally observed that photoreceptor R7 and R8, classically known for detecting colours [36], also exhibit orientation selectivity (Figure 3b). This unexpected finding raises the question of how such tuning arises in primary sensory neurons. We investigated the connectivity of the neurons and found that Dm9, R7 and R8 formed a closed recurrent loop that results in R7 and R8 adopting the same orientation preferences as Dm9 (Figure 1b and right two panels of Figure 3a), which itself receives input from L3. This closed recurrent loop between Dm9, R7 and R8, leading to synchronised orientation preferences, could provide new insights into how feedback within small circuits may amplify or stabilise neural responses. While this was not the primary hypothesis, it raises interesting questions about the role of recurrent circuits in orientation selectivity. Further exploration of this feedback mechanism in the future could contribute to our understanding of how local orientation selectivity is maintained or enhanced within neural networks.

## 4.2 Orientation maps

To evaluate whether orientation selectivity is spatially organised in the fly visual system, we focused on the medulla, a key visual processing region that receives direct input from L1-L3 neurons and contains the highest density of neurons with strong orientation selectivity (Figure 3b). Across the medulla sublayers - which are anatomically divided into distal (Dm), serpentine (Sm) and proximal (Pm) layers, shown in Figure 1a, we identified 1710 neurons in the Dm layer, of which 1245 belonged to the Dm3v, Dm3q and Dm3p subtypes. Additionally, we identified 143 well-fit neurons in the Pm layer and 63 in the Sm layer, based on the Gaussian tuning criteria (see Methods). It was found that neurons in all three sublayers exhibited clustered patterns - neurons with similar preferred orientations tended to cluster together (Figure S6). We then focused on the Dm and Pm layers for detailed spatial analysis, as these layers contained a sufficient number of neurons spanning the full spatial extent of the medulla (Figure S6).

We first visualised the spatial organisation of preferred orientation by flattening the 3D morphology of the neurons in the Dm and Pm layers onto layer-specific 2D coordinate systems. To achieve this, we computed a best-fit plane through the full set of 3D skeleton coordinates from each layer, providing a reference frame for spatial alignment and visualisation. This approach preserved the columnar layout while minimising distortions introduced by the *Drosophila's* optic lobe curvature. To enable cross-layer comparisons, a scaling transformation was applied to normalise for differences in physical size and curvature between the Dm and Pm regions. This projection preserves relative topographic relationships while enabling 2D spatial smoothing and angular comparisons. We colour-coded the neurons into their preferred orientation across Dm and Pm layers using unsmoothed maps to examine the spatial distribution of preferred orientations (Figure S6). These maps revealed a clear spatial clustering of similar orientation preferences, in contrast to a salt-and-pepper pattern in the rodent V1 area.

To better visualise the global structure, we applied local spatial smoothing by computing the circular mean of preferred orientations within a circular neighbourhood of fixed radius $r$ around each neuron. Specifically, for each neuron in the 2D projected plane, we identified all neurons located within a disk of radius $r \approx 5 \times 10^3$ nm and computed the circular mean of their orientation preferences. This smoothing preserved mesoscale spatial patterns while reducing local variability. The resulting heatmap produced orientation maps shown in Figure 4a. These smooth maps reveal pinwheel-like singularities - points around which orientation preference rotates continuously (the circled points in Figure 4a) - indicating a degree of spatial coherence previously unreported in invertebrate visual systems.

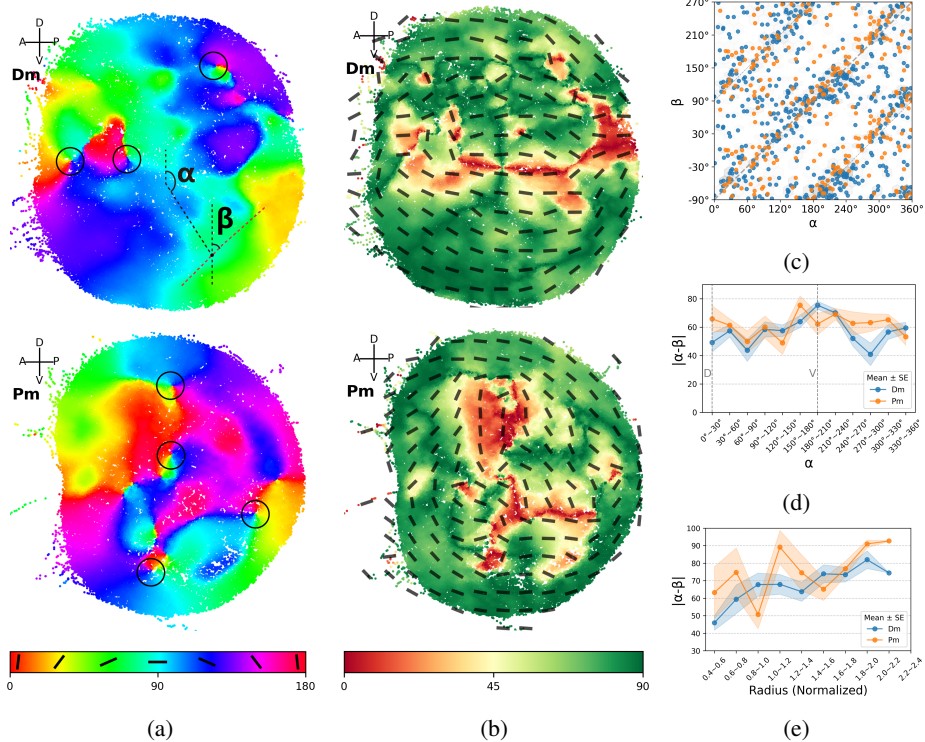

Figure 4: **Spatial and orientation map in the optic lobe.** (a) The smoothed heatmap of the optic lobe coloured by its preferred orientations at each layer of the optic lobe. The circles highlight the observed pinwheel structures. $\alpha$ denotes the position angle where dorsal-ventral represents $0°$ and $\beta$ denotes the preferred orientation of the neurons (see Methods). (b) Vector field plots of the preferred orientation in the Dm (top) and Pm (bottom) layers. Each line represents the preferred orientation of a neuron, and the background heatmap encodes the absolute angular difference between the neuron's $\alpha$ and $\beta$. Green regions indicate strong tangential alignment ($\Delta \approx 90°$), red regions indicate radial alignment ($\Delta \approx 0°$). (c) Scatter plot (position angle, preferred orientations) pairs are displayed with coordinate axes shifted for visualisation (y-axis: $-90°$ to $270°$, x-axis: $0°$ to $360°$). (d) Relationship of $|\alpha - \beta|$ and $\alpha$. D: dorsal area; V: ventral area. (e) Relationship between $|\alpha - \beta|$ and radial distance from the centre.

Initial observations of Figure 4a revealed that both layers exhibited a roughly centrosymmetric distribution of preferred orientations. To characterise this structure more precisely, we defined a polar coordinate system for each layer, with the origin set at the centroid of the neuron population, computed by the mean of the spatial coordinates, and using the V-D (bottom-top) direction as the axis. This allowed us to compute a spatial position angle $\alpha$ for each neuron and directly compare it to its preferred orientation $\beta$, derived from the smoothed map in Figure 4a. Building on this smoothed representation, we treated each neuron's orientation as representative of its surrounding neighbourhood, based on the same local averaging kernel described earlier. To quantify alignment, we computed the angular difference $\Delta = |\alpha - \beta|$ for each neuron and projected it onto a 2D heatmap (Figure 4b), where $\Delta = 90°$ indicates tangential alignment and $\Delta = 0°$ indicates radial alignment. This visualisation revealed a clear trend: neurons in peripheral regions tended to prefer tangential orientations ($\beta \approx \alpha + 90°$). These findings suggest that orientation tuning in the medulla is not randomly distributed, but spatially organised relative to each neuron's anatomical position.

We then further examined the relationship between $\alpha$ and $\beta$ for all neurons in Dm and Pm layers. Here, each neuron's position angle was computed as the circular mean of the angular positions of all its voxels in the 2D projection. The resulting scatter plot showed a clear diagonal band, consistent with a systematic offset where $\beta \approx \alpha + 90°$, supporting the presence of tangential alignment (Figure 4c). To quantify this relationship, we computed $\Delta$ and examined how it varied across spatial dimensions. Across angular bins, the $\Delta$ remains relatively stable with minor regional differences

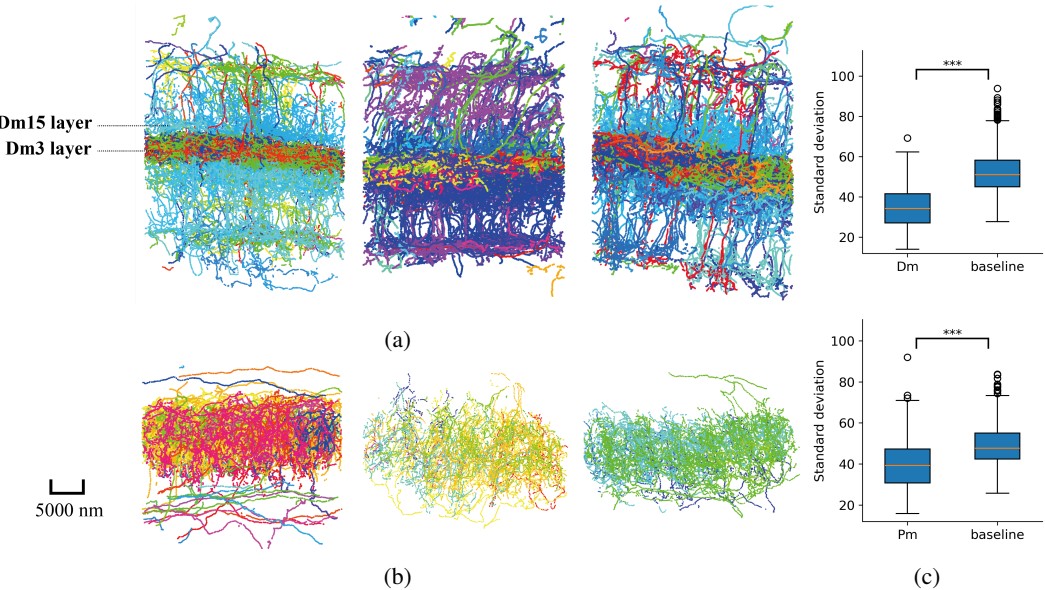

Figure 5: **Orientation column structure in the optic lobe.** (a) Three examples of orientation columns in the Dm layer, colour-coded by preferred orientation. (b) Three examples of orientation columns in the Pm layer, colour-coded by preferred orientation. (c) Quantification of tuning consistency within columns. Top: Dm layer. Bottom: Pm layer. *** indicates significance assessed via Mann-Whitney U test comparing standard deviations to baseline ($p < 10^{-20}$).

(Figure 4d). In contrast, alignment improved monotonically with radial distance from the centre (Figure 4e), indicating stronger alignment at the periphery. Although perfect tangential alignment would yield $\Delta \approx 90°$, the observed means approaching $90°$ suggest a systematic but approximate offset. We speculate that radial and tangential recognition neurons organise an overall pattern in the medulla corresponding to the horizontal direction. This pattern is especially evident in the Dm layer, suggesting a non-uniform organisation of orientation preferences, which may play a key role in the previously reported recognition of horizons and horizontal objects by the biological flight control system [47].

This finding suggests that orientation tuning in *Drosophila's* optic lobe is not randomly distributed but exhibits a structured, position-dependent organisation. To access the contribution of upstream visual pathways to orientation selectivity, we performed a targeted ablation analysis (Appendix C). Silencing Mi neurons abolished tuning across both Dm and Pm layers, while silencing Tm neurons produced selective loss in Pm but minimal effect in Dm, suggesting that Mi neurons provide essential input for orientation tuning throughout the optic lobe. Consistent with this observation, an analysis of synaptic connectivity by orientation preference (Appendix D) revealed that excitatory neurons with similar tuning are more likely to form recurrent connections, supporting topographic organisation and collinear facilitation as proposed by Seung [45]. In contrast, inhibitory connections displayed distinct off-diagonal structure, suggesting selective cross-orientation motifs that may sharpen tuning and maintain balance within the network. Peripheral neurons preferentially align their orientation selectivity tangentially relative to the centre of the optic lobe, reminiscent of contour-aligned representations seen in higher animals.

## 4.3 Orientation columns

We next asked whether such tuning is preserved across the depth of the medulla. We examined the existence of column structures - clusters of neurons aligned vertically across layers that share similar orientation preferences. Putative orientation columns were defined by anchoring each analysis region around a Tm3 neuron, chosen due to their well-defined columnar morphology. For each anchor neuron, we extracted the surrounding neurons within a cylindrical region of $1.5 \times 10^{4}$ nm radius for both the Dm and Pm layers according to their skeleton coordinates. Each such grouping was considered as a single column. A total of 858 columns were extracted, each containing on average

approximately 158.29 neurons in the Dm layers, of which 124.14 belong to Dm3 and Dm15 neuron types and 29.7 neurons in the Pm layers.

We visualised the neuron morphology by projecting a rectangular slice passing through the centre of the cylinder onto a 2D plane. This plane was defined by the two orthogonal axes perpendicular to the principal axis - the longest direction along which the neuron's morphology extends - of the Tm3 skeleton, ensuring that the projection aligned with the local column geometry. Specifically, neuron positions were transformed into this local coordinate frame by projecting each skeleton point onto the two orthogonal basis vectors to the Tm3 axis. This allowed us to flatten the 3D column into a 2D view while preserving vertical alignment across layers. We randomly selected three distinct columns each from Dm (Figure 5a) and Pm (Figure 5b) for visualisation. The Dm layer did exhibit orientation column organisations, with vertically clustered neurons sharing similar orientation preferences. However, Dm3 (Dm3v, Dm3p, Dm3q) and Dm15 neurons seemed to form a distinct horizontal layer that bisects each column into upper and lower segments, creating a clear visual stratification within individual columns (Figure 5a). This pattern was observed consistently across columns, indicating a widespread structural feature in the medulla. Notably, these neuron types exhibited only a single orientation preference (Dm3v: $\approx 0°$; Dm3p: $\approx 60°$; Dm3q: $\approx 120°$; Dm15: $\approx 100°$), in contrast to other types that exhibited broader tuning curves or gradual changes in preferred orientation across space (Figure 2a). On the other hand, the Pm layer also exhibited orientation column structure (Figure 5b), though there are a few columns that seem less pronounced in the structure (left two panels of Figure 5b).

To quantify this alignment, we measured the circular standard deviation (see Appendix B.4) of preferred orientations within each column in the Dm layer. The circular standard deviation was computed over the preferred orientations of all voxels within the column region, providing a measure of how tightly aligned the local tuning preferences are. Lower values indicated stronger orientation coherence within the column. To further ensure that the observed orientation columns were not an artefact of the simulation framework or analysis pipeline, we generated a null baseline by randomising the preferred orientations across neurons while preserving their spatial locations and column definitions. The resulting distribution revealed that most Dm columns, excluding Dm3 and Dm15, exhibited sharp tuning, with a peak at low deviation values and a minority displaying broader orientation spread (top panel of Figure 5c). We repeated the same analysis for columns in the Pm layer. In contrast to Dm, the Pm layer showed weaker columnar correspondence, with a distribution of standard deviations centred closer to the shuffled baseline (bottom panel of Figure 5c). Compared to the Pm layer, the Dm layer exhibited a more pronounced separation between the observed and baseline distributions, indicating stronger tuning coherence.

Together, these results demonstrate that *Drosophila's* optic lobe contains not only intra-layer orientation maps, but also orientation columns across different layers of Dm, reminiscent of columnar architectures in vertebrate visual systems.

# 5   Conclusion and Discussion

In this study, we investigated the orientation preferences in the *Drosophila* visual system and explored the spatial organisation of these preferences, revealing a structured pattern. While these maps are less discretely defined than in mammals, likely due to the lower neuronal density, the presence of spatially organised orientation selectivity and columnar structure suggests that *Drosophila* possesses a rudimentary form of orientation map. Biologically, the organisation shows a population-level bias toward tangential tuning, where neurons prefer orientations aligned with the local contour of the visual field and may reflect an adaptation for encoding object boundaries or motion, enhancing contrast sensitivity across the curved surface of the compound eye [6].

The results align with studies in other non-mammalian species, such as birds, turtles and zebrafishes, where sharply tuned orientation-selective neurons are found in non-laminated visual pathways as mentioned earlier [32, 17, 40, 43]. However, as of now, *Drosophila* is currently the only invertebrate where spatially structured orientation preferences, resembling a primitive orientation map, have been computationally demonstrated. As such, we hypothesise that structured visual maps may reflect convergent evolution, arising from shared computational demands rather than shared anatomy. The *Drosophila* ommatidium $\rightarrow$ Dm $\rightarrow$ lobula circuit may serve a functionally analogous role to the mammalian retina $\rightarrow$ LGN $\rightarrow$ V1 pathway [1, 19, 27], suggesting that efficient visual processing

follows common architectural motifs across species. Mechanistically, dendritic integration, closed-loop motifs, and lateral inhibition likely contribute to orientation tuning in *Drosophila*, as proposed in prior works [2, 25, 45]. Consistent with this interpretation, analysis of synaptic connectivity by orientation preference (Appendix D) revealed that neurons with similar tuning are more likely to form recurrent connections. The study by Klapoetke et al. [30] reveals functionally clustered representations of complex visual features in the lobular columnar neurons projecting into the central brain. In contrast, out work focuses on orientation selectivity in the earlier stages of the optic lobe, based on connectome-driven circuit structure. Together, these complementary findings highlight how structured visual maps may extend from early visual encoding to higher-order feature integration within the *Drosophila* brain.

The orientation maps at the early stages of visual processing provide critical advantages for efficient visual detection for the flies. The neurons that selectively respond to similar orientations are spatially adjacent to each other, enabling them to fire maximally when a line or edge appears at their preferred orientation, which in mammals is achieved by long-distance connections among columns with similar orientation preference. The columns are arranged in a primitive pinwheel-like map, enabling more stable and precise edge perception. This organisation supports higher-level vision by feeding processed orientation data to the central brain for shape recognition and motion detection. *Drosophila* serves as a powerful minimal model for exploring core principles of vision, with potential applications in bio-inspired machine vision and low-power autonomous systems.

**Limitations.** This study relies on biologically grounded simulations and lacks in vivo validation, warranting experimental follow-up to confirm our predictions. Similar to prior work based on anatomical reconstructions [45], our approach emphasised predictive modelling to inform future experiments. The use of a simplified LIF model omits nonlinear firing dynamics and neurotransmitter effects, which may affect network behaviour. However, similar simplified models have been effectively used in large-scale cortical simulations to gain insights into emergent network dynamics [46, 18].

## Acknowledgments and Disclosure of Funding

This work was supported by the National Key Research and Development Program of China under Grant 2021ZD0200301, the National Natural Science Foundation of China under Grant U2341228, the Fundamental and Interdisciplinary Disciplines Breakthrough Plan of the Ministry of Education of China under Grant JYB2025XDXM504, and Ant Group under Grant 20252000099.

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

# Supplementary Material

## A    Code and data availability

The full code and data can be found at `https://github.com/JNLiew/flylif_orientation_maps`. The model ran on an Intel(R) CPU at 2.90GHz machine using 20 threads in parallel. Each thread required approximately 72 hours to complete a single round of simulation of different angles in the range $0°$ - $180°$ with an interval of $0.4°$.

## B    Experimental and analysis definition

### B.1    LIF model design

We implemented a conductance-based leaky integrate-and-fire (LIF) model [9, 21, 26, 28, 31, 46] and integrated the FAFB dataset into this LIF model to run our simulations [8, 13, 14, 20, 33, 35, 36, 44, 50]. The membrane potential $v_i$ of neuron $i$ evolves as:

$$\frac{dv_i}{dt} = \frac{g_i - (v_i - V_{\text{resting}})}{T_{\text{mbr}}}, \tag{1}$$

$$\frac{dg_i}{dt} = -\frac{g_i}{\tau}, \tag{2}$$

$$g_i \leftarrow g_i + (w_{j,i} * w_{syn}) \quad \text{upon spike from neuron } j. \tag{3}$$

A spike is emitted when $v_i \geq V_{\text{threshold}}$, after which the membrane potential is reset to $V_{\text{reset}}$ for a refractory period $T_{\text{refractory}}$. Synaptic transmission is delayed by a fixed latency $T_{\text{dly}}$. The model used the following parameter values obtained from *Drosophila* modelling or electrophysiology efforts [9, 26, 28, 42, 46]:

- $V_{\text{resting}} = -52$ mV: resting potential [28, 46]
- $V_{\text{reset}} = -52$ mV: reset potential [28, 46]
- $V_{\text{threshold}} = -45$ mV: spiking threshold [28, 46]
- $R_{\text{mbr}} = 10\,\text{k}\Omega\,\text{cm}^2$: membrane resistance [28, 46]
- $C_{\text{mbr}} = 2\,\mu\text{F}\,\text{cm}^{-2}$: membrane capacitance [28, 46]
- $T_{\text{mbr}} = R_{\text{mbr}} \cdot C_{\text{mbr}}$: membrane time constant [46]
- $T_{\text{refractory}} = 2.2$ ms: refractory period [28, 31, 46]
- $\tau = 5$ ms: synaptic decay time constant [26, 46]
- $T_{\text{dly}} = 1.8$ ms: synaptic transmission delay [46]
- $w_{\text{syn}} = 1.5$ mV: synaptic weight; free parameter
- $g_i$: the synaptic conductance resulting from the aggregate firing of neurons presynaptic to neuron $i$.

$w_{\text{syn}}$ is a free parameter representing the strength of excitatory and inhibitory postsynaptic potentials. The value was chosen to ensure sufficient firing activity across the medulla for analysis, given the low baseline firing rates of visual neurons. The number of synapses of an upstream neuron $j$ on a downstream neuron $i$ is represented by the connectivity $w_{j,i}$ such that if neuron $j$ fires, the membrane potential of neuron $i$ changes in proportion to its connectivity $w_{j,i} \cdot w_{syn}$.

To assess sensitivity to parameter perturbations, we performed a variation analysis where all parameters (except $V_{rest}$ and $V_{threshold}$) were randomly perturbed up to a fixed maximum percentage (Figure S1a). At 10% variation, we observed minimal changes in preferred orientation across neurons (Figure S1b). At 50% orientation selectivity was noticeably degraded (Figure S1c; however, surprisingly, the ODEs (Equations 1 and 2) remained stable and did not diverge. This suggests that the system retains a degree of functional robustness even under large perturbations.

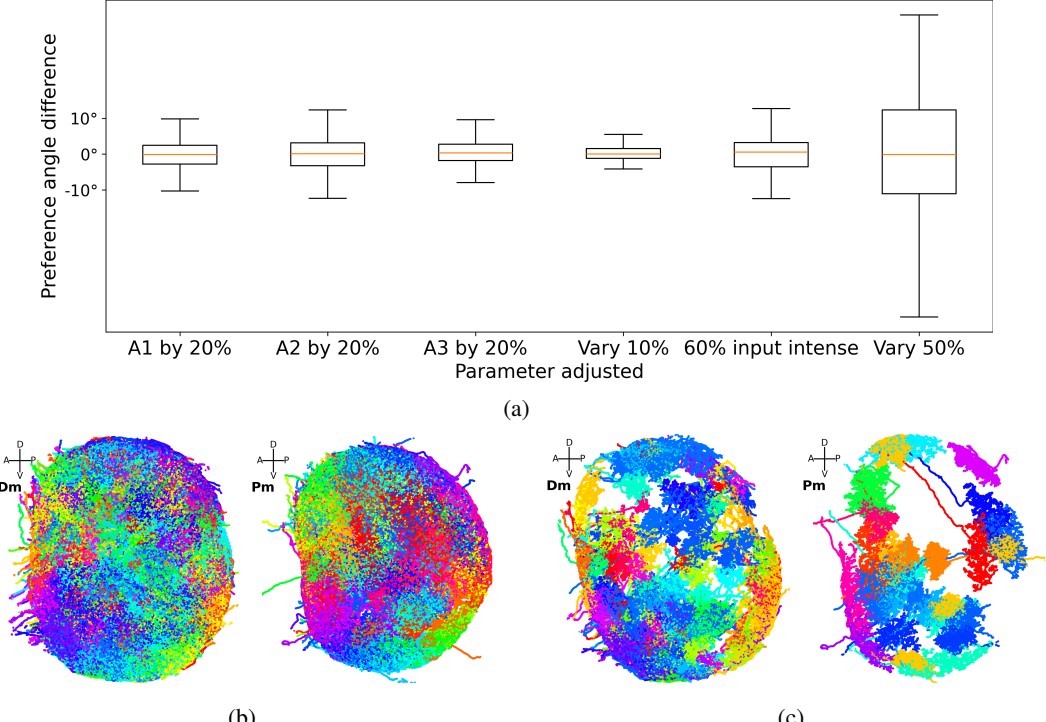

Figure S1: **LIF model parameter validation.** (a) Shows the difference in neurons' preferred orientation angles when stimulus parameters (A1, A2, A3) are increased by 20% respectively. "Vary 20%" indicates simulations where all LIF parameters (except $V_{rest}$ and $V_{threshold}$) were randomly perturbed by $\pm 10\%$. "60% input intense" refers to simulations where all inputs were reduced to 60%. "Vary 50%" is similar to "Vary 10%" but instead is randomly perturbed by $\pm 50\%$. (b) Orientation maps under 10% LIF parameters variation. (c) Orientation maps under 50% LIF parameters variation.

## B.2  Stimulus designs and validation

To simulate early visual input, we designed a synthetic stimulus that mimics the spatial layout and angular sampling properties of *Drosophila* photoreceptors. This allowed us to approximate the input that L1-L3 would receive in vivo. Our model design was guided by calcium imaging data reported in Ketkar et al. [29], which showed that L3 neurons exhibit the strongest response variation across changes in light intensity, followed by L1, with L2 showing the weakest modulation. These responses increased nonlinearly with light intensity, starting slowly and then rising more steeply. To capture this, we defined each neuron's firing rate as a function of spatial distance from a bar stimulus, which we refer to as the OFF-bar.

We assumed that spatial distance from the stimulus approximates variations in local light intensity across the visual field. The firing rate of neuron group L1-L3 was then defined as:

$$\text{FiringRate}_{Lx}(d) = \begin{cases} A_x \cdot \sqrt{1 - d^2} \cdot \frac{\max(fr)}{10}, & \text{if } |d| < 1, \\ B_x, & \text{otherwise}, \end{cases} \tag{4}$$

where $d$ denotes the distance from the centre of the ommatidium to the OFF-bar in the visual field, measured in units where $d = 1$ corresponds to the distance between the centres of two neighbouring ommatidia. $\max(fr)$ represents the maximum firing rate, set to 200Hz. The parameters $A_x$ and $B_x$, layer-specific gain and baseline levels, were estimated by fitting model responses to experimental data [29]:

- L1: $A_1 = 7$, $B_1 = 20$ Hz
- L2: $A_2 = 5$, $B_2 = 20$ Hz
- L3: $A_3 = 10$, $B_3 = 20$ Hz.

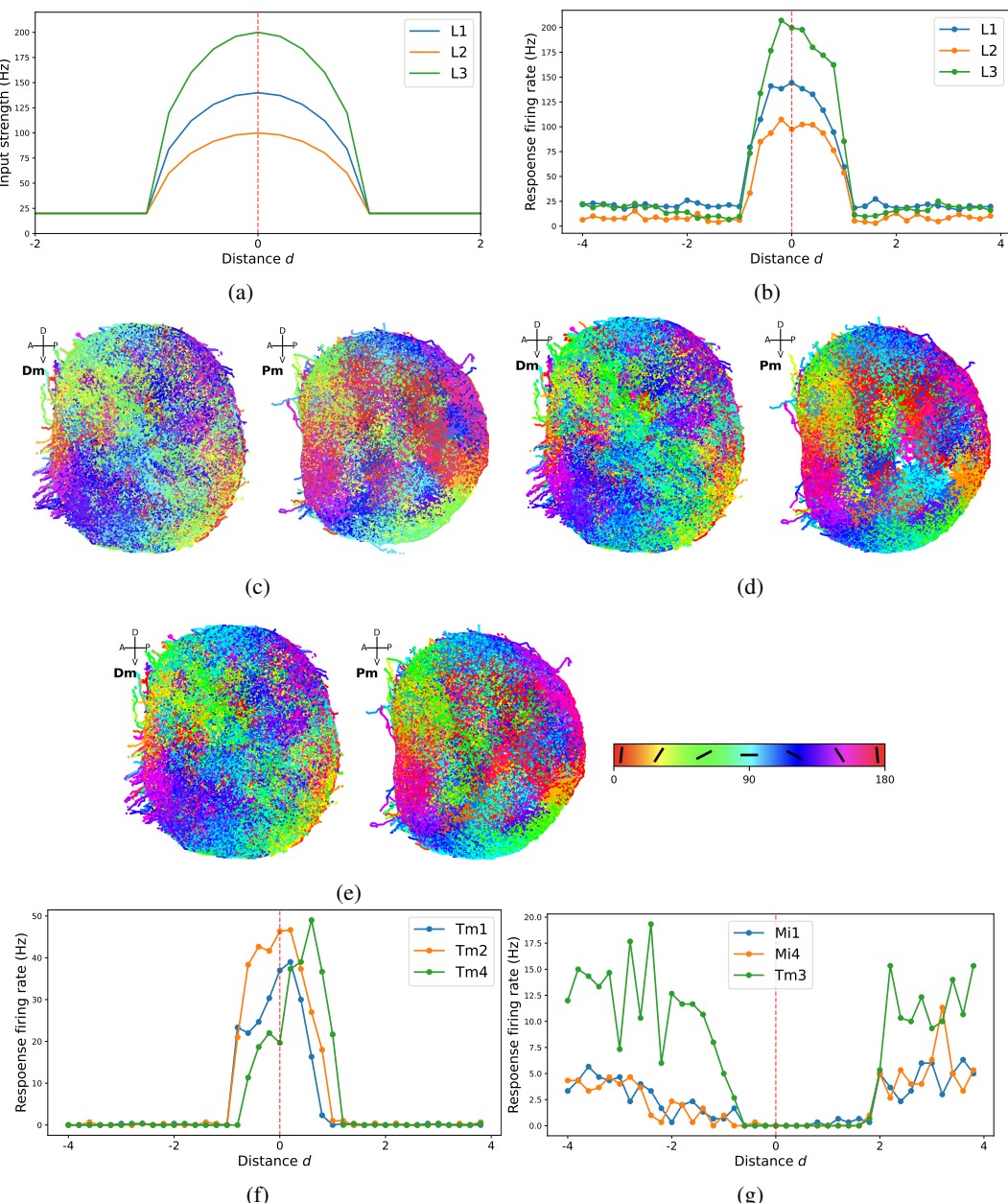

Figure S2: **Activation of neuron types to bar-like stimuli.** (a) Modelled input strength profiles for lamina neurons L1-L3 as a function of distance $d$ from the OFF-bar stimulus. (b) Firing rate traces of a representative neuron from each of the L1-L3 types against their distance from the OFF-bar. The red line indicates where the stimulus OFF-bar is located. (c) Orientation maps of where A1 parameters are varied by 10%. (d) Orientation maps of where A2 parameters are varied by 10%. (e) Orientation maps of where A3 parameters are varied by 10%. (f) Firing responses of a representative downstream OFF-response neuron from each of the Tm1, Tm2 and Tm4 types against their distance from the OFF-bar. In both (f) and (g), neurons were randomly sampled to verify that OFF-bar stimuli reliably activated these neuron types as expected. (g) Firing responses of a representative downstream ON-response neuron from each of the Mi1, Mi4 and Tm3 types against their distance from the OFF-bar. OFF-bar stimuli reliably suppress firing rates in these neurons within OFF-bar regions.

The configuration ($A_3 > A_1 > A_2$) captured the relative sensitivity of L1-L3, as shown in Figure S2a. The baseline parameter $B_x$ accounted for residual activity even when the calcium signal is low or flat, as $\Delta F/F = 0$ does not necessarily imply zero firing due to background fluorescence and imaging noise. We therefore set $B_x = 20$Hz to reflect plausible baseline activity. To assess the robustness of our results to the gain parameters $A_x$, we conducted a sensitivity analysis by decreasing each of $A_1$, $A_2$ and $A_3$ by 20%, one at a time. The changes in well-fit downstream neurons' preferred orientation (measured by preference angle difference) remained small across all conditions (Figure S1a), and the qualitative tuning properties were preserved. Similarly, we plotted the orientation maps of where A1, A2 and A3 parameters were varied. We observed minimal changes to the orientation selectivity (Figure S2c, S2d, S2e).

To ensure that the simulated stimulus was able to elicit a stimulus-consistent pattern among lamina, which is convinced to be a direct downstream of the photoreceptors, we examined the spatial and temporal activation patterns of lamina cells L1-L3 and their downstream neurons. Since our stimuli (OFF-bars) presented a straight bar pattern, we examined the firing rate of lamina cells when the OFF-bars were at different distances from them. We investigated the firing rate of lamina cells in response to the OFF-bars positioned at varying distances from them. Our findings revealed a clear trend: as the distance between the OFF-bars and the cell increased, the firing rate decreased correspondingly. Notably, the lamina cells (L1-L3) exhibited a peak firing rate when the OFF-bars were precisely aligned with the cell's location (Figure S2b). These results confirmed that artificial stimulus-driven lamina neuron activity in our digital *Drosophila* brain was consistent with the L1-L3 response properties known from biological experiments [29].

We then sampled a set of downstream neurons - specifically Tm1, Tm2, Tm4, Mi1, Mi4 and Tm3, which are known from anatomical studies to be postsynaptic targets of L1-L3 neurons [36]. These neuron types were not shown in Figure 1b to maintain visual simplicity, but their connectivity is well established. We randomly selected a single neuron from each of the Tm1, Tm2 and Tm4 subtypes as they are known OFF-cells [36] and evaluated if they exhibited firing patterns that reassemble the expected response profile to OFF-bar stimuli - that is, peaking when the bar aligns with their receptive field and decreasing with distance. As shown in Figure S2f, these OFF-responsive downstream neurons exhibited patterns as expected. In contrast, we also sampled a single neuron from each of the Mi1, Mi4 and Tm3 subtypes, known ON-cells [36], and evaluated whether they showed the opposite trend - stronger inhibition, when the OFF-bar was nearby (Figure S2g). These neurons' spatial activation pattern mirrors the OFF-bar's location on the compound eye, consistent with known visual topology [36].

## B.3    Circular Gaussian function

We calculated the preferred orientations of each neuron using a circular Gaussian function defined as:

$$\text{FiringRate}_n(\theta) = C \cdot \exp(-\frac{d(\theta, A)^2}{B}) + D, \tag{5}$$

where:

- $\text{FiringRate}_n(\theta)$: predicted firing rate of neuron $n$ at the orientation $\theta$
- $\theta$: input orientation (in degrees, $0° \leq \theta < 180°$)
- $A$: preferred orientation (in degrees, fit parameters)
- $B$: width parameters (sharpness of tuning)
- $C$: amplitude of the tuning curve
- $D$: baseline firing rate
- $d(\theta, A) = \min(|\theta - A|, 180 - |\theta - A|)$: smallest circular distance between input orientation and preferred orientation.

## B.4    Circular standard deviation

We quantified the sharpness of orientation tuning within each column using the circular standard deviation (see Figure 5c) in the `scipy` Python library, defined as:

$$s = \sqrt{-2 \ln R} \tag{6}$$

Table S1: Ablation study results.

| Z  | Tm silenced | Mi silenced |
|----|-------------|-------------|
| Dm | 90.7%       | 2.4%        |
| Pm | 10.4%       | 3.9%        |

where:

- $s$: circular standard deviation (in radians)
- $R$: mean resultant length, computed as:

$$R = \frac{1}{n}\sqrt{(\sum_{i=1}^{n}\cos\alpha_i)^2 + (\sum_{i=1}^{n}\sin\alpha_i)^2} \tag{7}$$

- $\alpha_i$: preferred orientation of neuron $i$ (in radians)
- $n$: number of neurons in the column

This metric captured how tightly clustered the orientations are around the mean. A value of $s = 0$ indicates perfect alignment, with larger values reflecting broader tuning distributions.

## C   Targeted ablation analysis

To evaluate the contribution of specific upstream visual pathways to orientation selectivity, we performed targeted silencing of key input neuron populations. Using the equation: $Z\% = X/Y$, where $X$ represents the number of good fits in ablation, $Y$ represents the number of good fits in the original result, and $Z$ is the percentage of neurons remaining that still exhibit orientation selectivity. Silencing Tm neurons resulted in an 89.6% reduction in well-fit orientation-selective neurons in the Pm layer but only a 9.3% reduction in Dm. In contrast, silencing Mi neurons eliminated tuning across both layers, leaving fewer than 5% of well-fit neurons in either Dm or Pm (Table S1). These findings suggest that Mi neurons represent essential upstream sources of orientation selectivity throughout the optic lobe.

## D   Orientation-based synaptic connectivity structure

To examine whether recurrent connections in the connectome reflect functional similarity, we quantified synaptic connectivity as a function of orientation preference. Each synapse $(p_i, q_i)$ was represented as a point $(\theta_{p_i}, \theta_{q_i})$, where $\theta$ denotes the preferred orientation (in degrees) of the pre- and postsynaptic neurons, respectively. In the excitatory subnetwork, connections cluster along the diagonal ($\theta_{p_i} \approx \theta_{q_i}$), indicating that neurons with similar orientation preferences are more likely to be recurrently connected (Figure S3a). This pattern is consistent with topographic organisation and colinear facilitation as proposed by Seung [45], supporting the notion of like-to-like connectivity in local circuits. In contrast, inhibitory connections exhibit off-diagonal structure, suggesting cross-orientation interactions and selective suppression mechanisms (Figure S3b). These results suggest that recurrent circuits in the *Drosophila* optic lobe preferentially link neurons with similar selectivity, providing network-level evidence for structured functional organisation within the connectome.

## E   Extended figures

Figure S4 shows the Gaussian model fitting used to quantify orientation selectivity. Figure S5 shows the orientation tuning profile of T4 and T5 subtypes. Figure S6 presents additional visualisation of orientation maps across different neuron types and layers within the optic lobe.

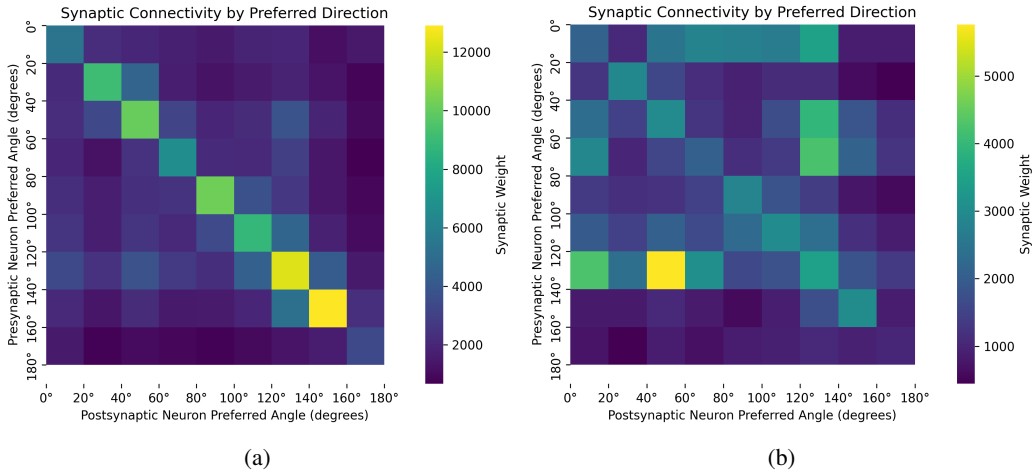

Figure S3: **Orientation-specific synaptic connectivity.** (a) Excitatory network showing strong diagonal structure, indicating neurons with similar orientation preferences preferentially connect. (b) Inhibitory network showing off-diagonal structure, consistent with cross-orientation suppression motifs.

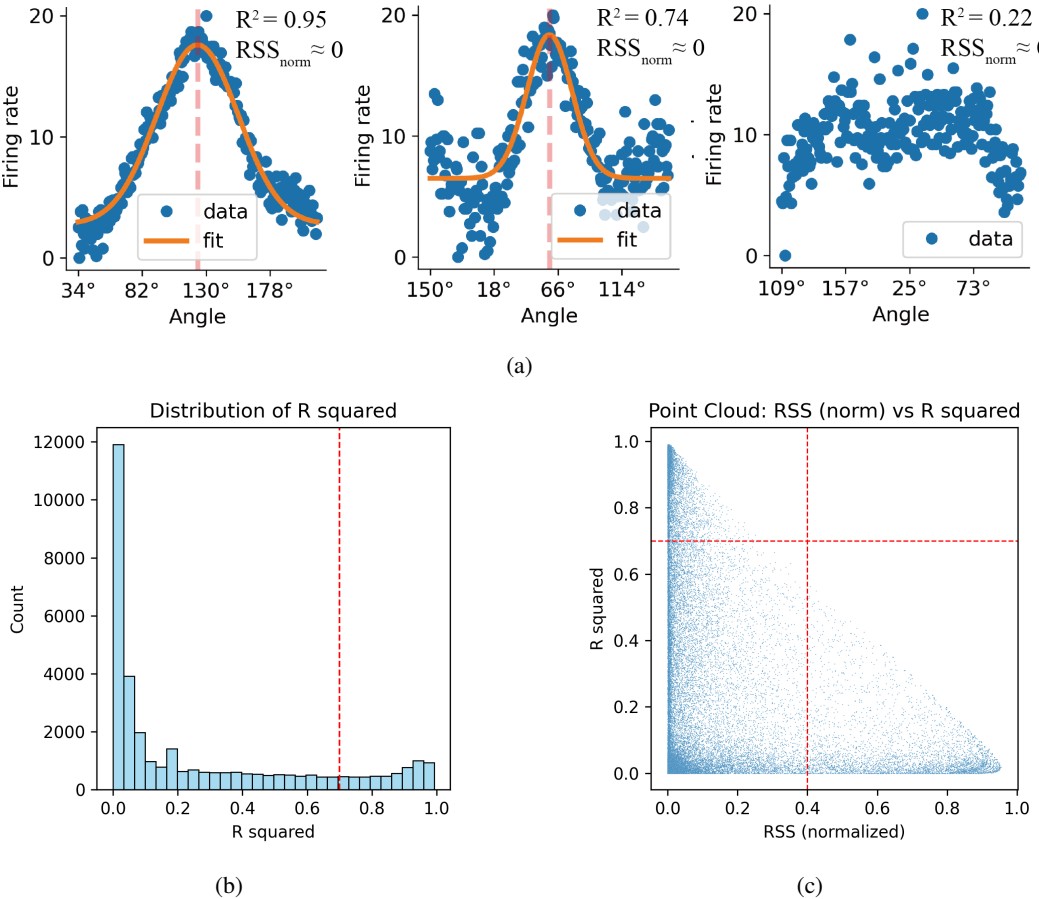

Figure S4: **Model Gaussian fit quality.** (a) Representative neurons where their Gaussian fit is considered good (left), near borderline (middle) and poor (right). (b) Distribution of $R^2$ values, where the red line indicates the threshold. (c) Distribution of $R^2$ and normalised $RSS_{norm}$ of all neurons in the right optic lobe, including neurons in Figure S4a.

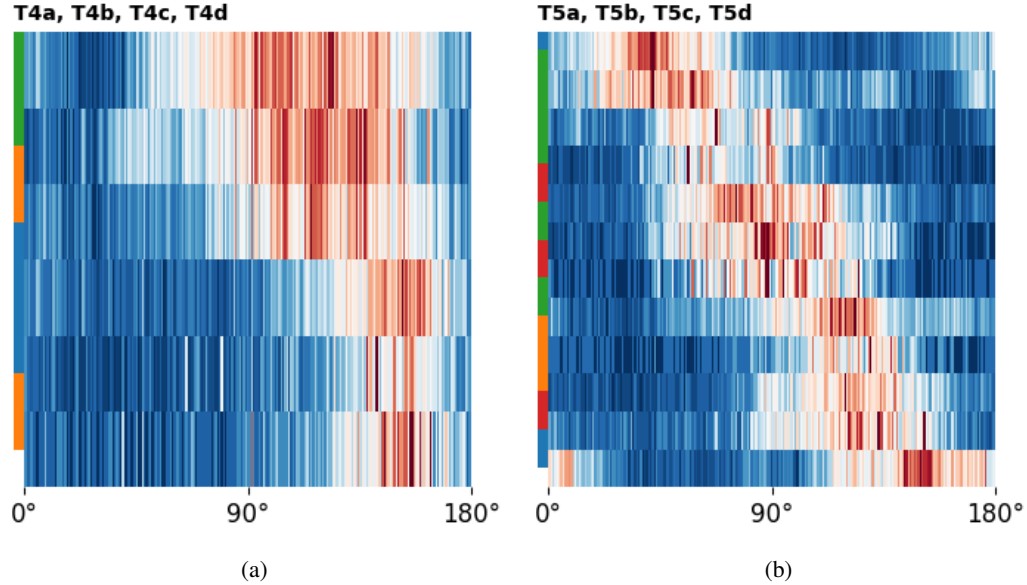

(a)              (b)

Figure S5: **Orientation tuning profiles of some T4 and T5 neurons.** (a) Heatmap showing the orientation tuning responses of T4 neurons that are well-fitted. (b) Heatmap showing the orientation tuning responses of T5 that are well-fitted.

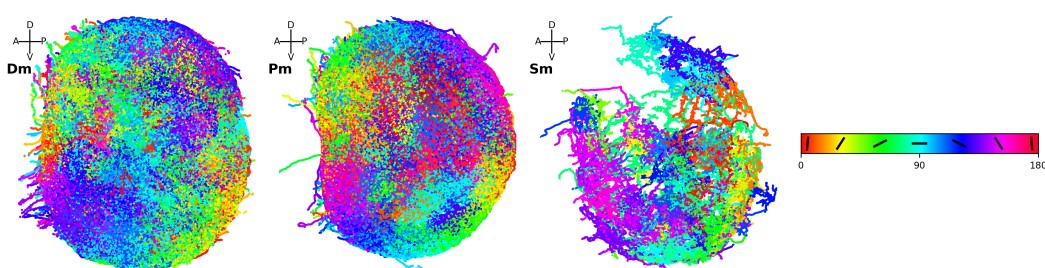

Figure S6: **Orientation preference structure across different neuron types.** Preferred orientation maps across optic lobe layers before smoothing, colour-coded by angle.

