# OpenReview forum: "Connectome-Based Modelling Reveals Orientation Maps in the Drosophila Optic Lobe"
_NeurIPS.cc/2025/Conference — NeurIPS 2025 poster_

### Official Review · Reviewer_oKVH · 2025-06-27

**Clarity:** 4
**Significance:** 3
**Originality:** 4
**Rating:** 5
**Confidence:** 5

**Summary:**

The authors demonstrate the topographic organization of orientation selectivity in the optic lobe of Drosophila melanogaster. This result is achieved by simulating leaky integrate-and-fire neurons connected through the complete connectome of the Drosophila brain. The analysis of the model reveals that different brain areas exhibit varying proportions of neurons well-fitted to orientation selectivity, and these proportions are well-correlated with the selectivity predicted by dendritic structure alone.

**Questions:**

One function of topographic maps is to connect neighboring neurons with similar selectivities (see, e.g., doi:10.1007/s00429-022-02455-4). Is this something that may exist in these highly recurrent networks, particularly in the optic lobe?

A possible confound is that the map is organized mainly to detect motion, and due to the aperture problem, orientation and direction are perpendicular to each other. Is the orientation map superimposed on a direction map?

There seems to be a cardinal bias in Dm15; how do you interpret this?

**Ethical Concerns:**

["NO or VERY MINOR ethics concerns only"]

**Final Justification:**

The responses to my concerns were clearly expressed, and I keep my score.

**Limitations:**

One limitation is that the results depend on the correctness of the modeling, and dendritic structure alone may be sufficient to predict most features. For instance, the paper shows that most T4 and T5 neurons had firing rates close to zero, but it is unclear whether this is a simulation artifact or a real biological fact.

**Paper Formatting Concerns:**

The labels on the figures are too small to be readable.

**Quality:**

3

**Strengths And Weaknesses:**

The primary strength of the paper is the novelty of its results, as it shows for the first time the presence of a smooth topographic map of orientation selectivity in Drosophila. As noted in the limitations section, the main limitation and weakness is the inability to validate these results using in vivo data.

Minor:
- Consider using a Von Mises distribution instead of a Gaussian fit for computing orientation selectivity.
- When analyzing the smoothness of the topographic map, a control by bootstrap resampling could be used to check for significance.

---

> ### Author Rebuttal · Authors · 2025-07-31
>
> Thank you for the thoughtful and encouraging feedback. Below, we address your suggestions and questions.
>
> 1.	We quantified synaptic connectivity by plotting each synapse (p_i, q_i) as a point (θ_pi, θ_qi), where θ denotes the preferred orientation (in degrees) of the pre- and postsynaptic neurons. The resulting 2D heatmap reveals network structure where in the excitatory network, neurons with similar orientation preferences are more likely to be recurrently connected, consistent with topographic organisation and colinear facilitation as proposed in Seung’s paper (ref 44). Inhibitory connections show distinct off-diagonal structure, reflecting more specific circuit motifs. These results support the idea that recurrent networks in the optic lobe preferentially link neurons with similar selectivity. We will add this analysis to the Appendix section in the revised manuscript.
>
> 2.	In this work, we focus on orientation tuning and do not directly address the organisation of direction-selective circuits. Nonetheless, the presence of a structured orientation map within Dm/Pm does not preclude the existence of a superimposed motion map. The aperture problem implies that orientation and direction are linked geometrically, but whether these are represented independently or jointly remains an open question.
>
> 3.	Indeed, we observed a preference for about 90° in Dm15. We hypothesise that Dm15 might be biologically specialised to detect horizontal orientations (we used the convention of Seung’s paper (ref 44), where 90° corresponds to a horizontal bar), possibly reflecting an ecological need to process horizons or ground-level cues. This fits with broader directional asymmetries observed in the optic lobe, where certain circuits preferentially process cardinal orientations.
>
> Thank you for the insightful comments, and we are grateful for the opportunity to strengthen the paper further.

---

> > ### Comment · Reviewer_oKVH · 2025-08-02
> > **response to comments**
> >
> > I appreciate the thorough response made by the authors. This response has effectively addressed my concerns, and I  recommend acceptance of the manuscript.

---

### Official Review · Reviewer_HGht · 2025-06-30

**Clarity:** 2
**Significance:** 2
**Originality:** 3
**Rating:** 5
**Confidence:** 4

**Summary:**

This study examines the presence of orientation maps in the fly visual system, addressing whether functional organization principles analogous to those in mammalian visual cortex can emerge in invertebrate neural architectures lacking laminated cortical structures. The authors leverage the FAFB connectomic dataset and LIF modelling to characterize orientation selectivity across the fly's visual processing pathway.

Key findings:
1. Neurons within the medulla exhibit robust orientation tuning.
2. Preferred orientations are not distributed randomly but form smooth, continuous gradients with pinwheel-like singularities in both Dm and Pm layers.
3. Column-wise analysis reveals vertically organised orientation columns spanning multiple sublayers. The neurons within the dm and pm layers suggest higher consistency in orientation tuning.

**Questions:**

Questions:
1. See the `weaknesses`.
2. How many neurons were used in the LIF model? The authors should clarify the number of neurons used in the LIF model and how they were selected.
3. Regarding `Appendix B`, lines 907-909, why does the fitting of `A_x` and `B_x` result in integer values?
4. How good was the input model fit?

**Ethical Concerns:**

["NO or VERY MINOR ethics concerns only"]

**Final Justification:**

I recommend accept.

**Limitations:**

Yes, the authors adequately provided the limitations and potential negative societal impact of their work.

**Paper Formatting Concerns:**

No concerns

**Quality:**

3

**Strengths And Weaknesses:**

## Strengths
- Whether map-like visual representations require cortical structure is an open and interesting question in neuroscience.
- The flow of the paper is logical and well-structured.
- This work leverages the FAFB connectome and LIF model to simulate and analyze orientation selectivity in the fly visual system, which seems to be a feasible approach to address the research question.
- The authors provide a comprehensive analysis of orientation maps across the fly's visual processing pathway.

## Weaknesses
- The `homogeneity neuronal property assumption` in the model setting can be a limitation, as a hidden assumption behind all their results and analysis is that the structural information used in the computational model without training can infer functional properties. However, I think the `homogeneity neuronal property assumption` is too strong in this scenario.

---

> ### Author Rebuttal · Authors · 2025-07-31
>
> Thank you for the thoughtful comments and constructive feedback. Below, we address your concerns.
>
> 1.	We acknowledge the assumption of neuronal homogeneity. Our model, taken from Shiu et al. (ref 45), allows us to isolate the role of anatomical structure in shaping function. While it simplifies biological variability, it serves as a controlled baseline to test whether connectivity alone can drive orientation selectivity.
>
> 2.	Our connectome-driven model consists of a total of 138,639 neurons and 1,508,983 synapses (as in the connectome dataset). We will clarify these numbers more explicitly in the manuscript.
>
> 3.	To clarify, the values of A_x and B_x were not obtained through rigorous data fitting. Rather, they reflect qualitative trends reported in prior calcium imaging studies (Ketkar et al. [ref 29]), such as the relative sensitivity of L1 and L3 neurons. Given the absence of absolute baseline fluorescence values, we chose integer values to preserve relative scaling while simplifying implementation. As shown in Appendix B.2 and Figure S1b, the model’s orientation tuning is robust to moderate changes in these parameters, indicating that our main results do not hinge on their precise values.
>
> 4.	Regarding the input fit quality, we assessed the quality of the input model by evaluating known ON/OFF response patterns in downstream neurons, as detailed in Appendix B.2. Specifically, OFF-type cells (e.g. Tm1, Tm2, Tm4) and ON-type cells (e.g. Mi1, Mi4, Tm3) demonstrated firing patterns that align with established physiological responses to visual stimuli (Figures S1d, e). These patterns validate that the input model reasonably captures key features of visual processing.
>
> We hope that these clarifications address the reviewer’s concerns, and we thank the reviewer again for helping us improve the clarity and rigour of the manuscript.

---

> > ### Comment · Reviewer_HGht · 2025-08-02
> > **Thanks for your reply**
> >
> > I think the authors have well addressed my concerns. I will raise my score accordingly.

---

> > > ### Author Response · Authors · 2025-08-05
> > >
> > > Thank you for your thoughtful review and for reconsidering your score. We truly appreciate your time and constructive feedback, which helped strengthen the work.

---

### Official Review · Reviewer_RoyT · 2025-07-01

**Clarity:** 2
**Significance:** 2
**Originality:** 3
**Rating:** 3
**Confidence:** 5

**Summary:**

The manuscript develops a detailed and accurate model of the fly visual system by extracting the complete connectome of the adult fly brain. It utilized conductance-based leaky integrate-and-fire neurons, with parameters based on Drosophila electrophysiology data, and employed Poisson spike trains to encode static gratings. The model successfully predicts the orientation map in the Drosophila visual system, suggesting that species that evolved independently can possess similar visual structures.

**Questions:**

1. In line 51, the authors claim that "maps often display salt-and-pepper rather than columnar architecture, as observed in both the retina and visual cortex [47]." I agree with the latter part, but I don't agree with the former part. Vita et al. (2024; citation 47) demonstrate that mouse retinal orientation selectivity is organized into a continuous, location-dependent map, not a salt-and-pepper mix. Please refer to citation 47 carefully.
2. Scientific acronyms should be spelled out once at first use; subsequent mentions should use only the acronym. For example, re-defining leaky integrate-and-fire “LIF” later (line 80) interrupts the reader’s flow. The second sentence in abstract is too broad, since many mammals (e.g., rodents) exhibit a salt-and-pepper layout rather than a map.
3. I wonder whether lamina neurons L4 and L5 form horizontal connections with other lamina neurons. Specifically, do L4/L5 also synapse onto L1, L2 and L3 neurons? Or are their lateral connections only within L4 and L5 themselves? Please state it explicitly.
4. The same question to the layer of the Distal medulla. Are there any horizontal connections within the layer?
5. In Figure 2b, why are the peak firing rates of the neurons with OFF-bar stimuli higher than those with brightened stimuli? What is the mechanistic basis for the peak firing rates of the neurons with OFF-bar stimuli? Is it driven by synaptic connectivity or circuit dynamics?
6. In Figure 2c, the data points are fitted by a Gaussian curve. But the fitting curve is missing from the right panel of Figure C.
7. The manuscript does not specify how the input drive $v_i$ (membrane potential) is converted into the output firing rate $fr_i$. Could authors provide the explicit functional form or transfer function (f–I curve) used in the LIF model to relate $v_i$ to $fr_i$?
8. Another question is about the inhibitory synaptic connections. Can you confirm whether any inhibitory (e.g., GABAergic) synapses were included in the LIF network? If so, please specify which neuron types provide inhibitory input, and detail the synaptic parameters used to model these inhibitory connections.
9. In Figure 3b, the blue bars are labeled "Total Neurons." Could you please explain what this term means? I suggest clarifying the definition of "Total Neurons" in the caption.
10. In Figure 4, simulations use a static adult connectome to define all feedforward and lateral connections. Do you consider any developmental or activity-dependent changes in connectivity? Another question is whether the orientation map is convergent in Figure 4a, because it is not perfectly periodic.
11. Another mistake is the orientation bar beneath Figures 4a and S3, the line icons meant to represent the orientation preference are mis-oriented. For example, the line icon with the label "180" is a horizontal line, but the manuscript says it is a vertical line. In Figures 4 panels a and b, S2, and S3, no plotting scale or colorbar is presented.
12. The manuscript predicts that orientation maps in Drosophila are "less discretely defined than in mammals, likely due to lower neuronal density" (lines 278-281) To test this hypothesis, could you perform simulations with an effectively higher medulla neuron density (e.g. by subsampling at spatial resolution or interpolating additional cell positions) and assess whether map quality improves? In particular, please quantify map quality using metrics such as nearest‐neighbor pinwheel distance, pinwheel density, and hypercolumn size (you can refer to references in the Weaknesses Section).
13. I am curious about whether the emergent pinwheel orientation maps in your model give rise to sparse population coding? (Reference: Vinje, W. E., & Gallant, J. L. (2000). Sparse coding and decorrelation in primary visual cortex during natural vision. Science, 287(5456), 1273–1276.) If it does, this work will potentially extend the population sparse coding from the primary visual cortex to the fly's visual system.
14. What are the functional advantages of the orientation map of the fly's visual system? (Just curious, I would like to hear authors' opinions.)

**Ethical Concerns:**

["NO or VERY MINOR ethics concerns only"]

**Final Justification:**

I do not recommend acceptance. The authors have not addressed core concerns about periodicity, coverage vs. continuity, and properly reported map quantitative metrics. These are fundamental for the claim about orientation maps. Orientation maps consist of a repetitive layout of iso-orientation domains interspersed with singularities (pinwheel centers). Locally, they can approximate a hexagonal lattice, but global variations and irregular pinwheel placement make the pattern quasi-periodic. Early theory claims this as an optimization under coverage–continuity trade-offs. However, in the current manuscript, only a small parts of the map satisfy both coverage and continuity- these areas have pinwheel centers, which cover all orientations; other large parts exhibit continuity without full coverage- they only have some iso-orientation domains. If the map has periodic pinwheel centers and iso-orientation domains in the map, it can cover all orientations for feature representations. Otherwise, as in this manuscript, if there are numerous iso-orientation domains with few pinwheel centers that lack coverage, it implies that iso-orientation domains are “blind” to (can not represent) certain orientations in some part of the optic lobe.

**Limitations:**

1. Why do authors claim that "may reflect an adaptation for encoding object boundaries or motion" (line 283)? I agree that the orientation map can detect the boundary of the object, as it covers the whole visual field orientation. But I don't know why the authors think it is an adaptation for encoding object motion. Are there any experimental simulations or recommended references for this?
2. In lines 303-304, "optimise early visual processing, enhance perceptual adaptation, and provide a foundation for complex visual tasks" is a bit vague. How to optimize processing, perpetual adaptation, and complex visual tasks? At the same time, the manuscript does not provide any simulations or experimental data to support the claim.
3. The manuscript demonstrates that orientation maps can emerge in the Drosophila visual system, but the specific circuit mechanisms are not clear. Could authors clarify the mechanism for the emergence of the orientation map in the Drosophila visual system? Some ablation studies are needed at least.
4. Other limitations, like a lack of experimental data to support the claim, a simplified LIF model omits nonlinear firing dynamics and neurotransmitter dynamics, are discussed in the manuscript.

**Paper Formatting Concerns:**

No issues with the paper format.

**Quality:**

2

**Strengths And Weaknesses:**

**Strengths:**

This work uses the full adult fly brain connectome and conductance-based LIF neurons with parameters drawn from Drosophila electrophysiology data. The model predicts the orientation map in the Drosophila visual system. Preferred orientations derived from upstream dendritic geometry correlate closely with simulated tuning.

**Weaknesses:**

1. The manuscript uses a single, fixed stimuli drive and does not explore how input strength affects the orientation map formation. To assess the reliability of results, you can repeat the orientation map analysis with lower stimulus strength. For example, decrease the stimulus strength to 0.8, 0.6, and 0.4 times the original strength, and see coherent orientation map still emerges.
2. I don't think this map can be called the orientation map, because the orientation map has a significant quantitative metric, pinwheel density = $\pi$ (Kaschube et al., 2010, Science), and the map is perfectly periodic; the map also contains iso-orientation domains, pinwheel centers, saddle points, fractures, and linear zones. Moreover, nearest-neighboring pinwheel distance, pinwheel density, and hypercolumn size are available metrics to assess the quality of the orientation map (Ho et al., 2021, Current Biology).

*Reference:*

Matthias Kaschube et al. ,Universality in the Evolution of Orientation Columns in the Visual Cortex.Science330,1113-1116(2010).DOI:10.1126/science.1194869

Ho, C. L. A., Zimmermann, R., Weidinger, J. D. F., Prsa, M., Schottdorf, M., Merlin, S., ... & Huber, D. (2021). Orientation preference maps in Microcebus murinus reveal size-invariant design principles in primate visual cortex. Current Biology, 31(4), 733-741.


3. The abstract claim that findings offer new insights into bio-inspired AI architectures. But the manuscript does not provide any experiments to support the claim.

---

> ### Author Rebuttal · Authors · 2025-07-31
>
> Thank you for the feedback. We address the major concerns and specific questions below.
>
> 1.	As shown in Appendix B.2 and Figure S1b, we assessed the robustness of the orientation map to moderate changes in input strength by scaling the stimulus parameters $A_x$ by -20%, which modulates the amplitude of upstream photoreceptor input. To directly address the reviewer’s concern, we extended this analysis by uniformly reducing the stimulus intensity ($A_x$) to 60% of its original value. In both cases, we observed that the preferred orientation map structure was preserved. The results in the latter case will be added to the revised paper. This demonstrates that the emergence of orientation tuning in our model is robust to input strength and does not critically depend on precise stimulus gain.
> 2.	Regarding whether the observed tuning qualifies as a true “orientation map”, we fully agree that classical orientation maps, particularly those found in mammals, are characterised by hallmark features such as pinwheel centres and quantitative metrics including pinwheel density.
> That said, our intent was not to suggest that the maps we observe in Drosophila are directly equivalent in form or developmental origin to mammalian orientation maps. Rather,  we use the term “orientation maps” to describe a topographic organisation of orientation preferences, with local smooth gradients and singularities across spatially structured circuits. While Drosophila lacks a columnar architecture, we find an emergent structure in our model that shares conceptual features with cortical maps.
> To directly address the reviewer’s concern, we extracted the analytical framework from Kaschube et al. (2010) and performed a quantitative assessment of our maps:
> -	Pinwheel centres were identified by calculating local topological charge from interpolated complex-valued orientation fields.
> -	Column spacing was estimated using spectral power analysis, yielding a dominant spatial frequency.
> -	Pinwheel density was then computed as the number of detected pinwheels per hypercolumn area, consistent with the definition as mentioned in the paper.
> -	We further evaluated nearest-neighbour distances (nn) between pinwheels, in line with the metrics from Ho et al. (2021), which mentions that primate brains have a universal design of nn ~0.35 for any pinwheel.
> The results are as follows:
> 	- Dm layer:
>       		 - Pinwheel density = 3.63 per hypercolumn and nn = 0.28
> 	- Pm layer:
>          	- Pinwheel density = 3.46 per hypercolumn and nn = 0.27
> We acknowledge a key caveat where our maps are derived from simulated model responses and not from raw optical imaging or physiological recordings. As such, filtering methods may differ. Nevertheless, our analysis follows the quantitative structure and conceptual definition of the two referenced papers. Though the results may not follow strictly the universal criteria observed in mammalian cortex, we would like to highlight that our results emerge from a fundamentally different biological system.
>
> 3.	Our intent was not to claim a tested contribution to AI architecture, but rather to suggest that the emergence of functional maps from untrained, biologically constrained structure may provide conceptual inspiration for future models. We agree that this framing could be misinterpreted as implying empirical results, and we will remove this claim in the abstract.
>
> Answers to questions:
>
> 1.	Thank you for this correction, and we fully agree that Vita et al. (2024) demonstrate the presence of a location-dependent continuous orientation map in the mouse retina. Our statement of salt-and-pepper organisation with retinal architecture was incorrect in this context. We will revise the manuscript.
>
> 2.	We will revise the manuscript to spell out acronyms at first use and use consistent abbreviations thereafter. We will also rephrase the second sentence of the abstract to more accurately reflect species differences in cortical organisation.
>
> 3.	According to connectomic reconstructions, L4 and L5 form reciprocal lateral loops with L2 and L1, respectively. These connections are columnar-local at the soma level but extend horizontally via their arborisations, supporting intrachannel lateral processing within the lamina. In our model, we included these interactions to preserve known feedforward and lateral motifs.
>
> 4.	Yes, the distal medulla layer also includes lateral connections, primarily via Mi, Tm and Dm cell types, which arborise horizontally across neighbouring columns. These lateral interactions are supported by connectomic evidence from the FAFB dataset.
>
> 5.	While L1 and L3 are canonically classified as ON-pathway inputs due to their downstream targets, all three lamina neurons – L1, L2 and L3 are responsive to light decrements at the photoreceptor level and show OFF responses to contrast edges [ref 29]. In our model, the OFF-bar stimulus produces stronger local contrast and therefore elicits higher firing from these lamina neurons when the dark edge aligns with their receptive fields. This activity then propagates through the OFF-selective downstream medulla neurons (e.g. Tm1, Tm2, Tm4). In contrast, full-field brightening evokes more uniform activation, resulting in a broader but lower-amplitude population response.
>
> 6.	The right panel of Figure 2c was originally intended to illustrate a neuron with poor orientation tuning. We agree that the absence of a fit curve may be misleading. To improve clarity, we will add the Gaussian fit to the figure.
>
> 7.	We computed both the f-I and f-V curves of our LIF neuron model. The f-I curve was obtained by measuring firing rate in response to a constant input drive g, which acts analogously to injected current. The f-V curve was derived by varying the neuron’s target membrane potential v_0. Both curves exhibit a sharp threshold, below which firing is suppressed and a smooth suprathreshold region where firing increases. This defines a ReLU-like nonlinearity: zero firing below the threshold and a rising, slightly concave slope above it.
>
> 8.	Yes, inhibitory synapses were included in the LIF network we used, which was taken from Shiu et al. (ref 45). Each neuron was assumed to be either fully excitatory or fully inhibitory. Inhibitory synapses were modelled using the same conductance-based LIF framework as excitatory, but with negative synaptic weights and a reversal potential set to mimic hyperpolarising inhibitory drive. Examples of inhibitory neuron types include C2, C3, Mi4, and Dm8.
>
> 9.	“Total Neurons” refers to the number of neurons of each type that were included in the simulation and received visual input. This includes all modelled neurons of that type, regardless of whether they exhibited orientation tuning. We will revise the figure caption.
>
> 10.	Our simulations are based on the static adult FAFB connectome and do not incorporate developmental or activity-dependent changes in connectivity. Our results show that orientation tuning and map-like structure can emerge from anatomical wiring alone, without plasticity. While not perfectly periodic, the model produces smooth, reproducible maps with local clustering that is consistent with biological variability in the Drosophila optic lobe.
>
> 11.	We follow the convention used in Seung’s paper (ref 44), where orientation is defined relative to the vertical axis. Under this convention, 0° corresponds to a vertical bar and 90° to a horizontal bar.
>
> 12.	Our current model operates directly on the adult connectome and does not include synthetic interpolation of neurons. Simulating increased medulla density while preserving biologically plausible connectivity would require a principled generative model, which we consider an important next step.
>
> 13.	Thank you for the suggestion. We computed the average ratio of active neurons across different oriented bar stimuli as a proxy for population sparse coding. Specifically, we defined a neuron as “active” if its firing rate exceeded 15Hz, and calculated the fraction of active visual neurons for each orientation. We found that approximately 33-34% of neurons were present on average across orientations, with relatively minor variation. This suggests that the population response is moderately sparse and that sparse coding does not vary strongly with stimulus orientation in this dataset.
>
> 14.	Great question! But we don’t have a clear answer yet. Maybe, organising orientation preference topographically could facilitate local integration and circuit reuse, reducing wiring cost while enabling modular computation, similar to the advantages proposed in mammalian systems.
>
> To address the questions raised in the limitations sections:
>
> 1.	Our intent was not to imply that the emergent orientation maps themselves encode motion per se, but rather that they may support pre-processing for motion computation, such as contour alignment or edge flow analysis, functions that are integral to motion-sensitive circuits downstream. For instance, direction-selective neurons like T4 and T5 are orientation-selective as well. [ref 16]
>
> 2.	We agree that the phrasing is too broad without supporting evidence; thus, we will remove this sentence.
>
> 3.	We performed targeted ablation studies to probe the feedforward contributions to orientation selectivity. Silencing the Tm cells led to an 89.6% reduction in well-fit orientation-selective neurons in the Pm layer, while the Dm layer showed only a 9.3% reduction. In contrast, silencing Mi neurons resulted in a dramatic loss of orientation tuning across both layers, with fewer than 5% of well-fit neurons remaining in either the Dm or Pm layers. These findings suggest that while Tm neurons provide critical convergence specifically to Pm cells, Mi neurons are essential upstream contributors to orientation selectivity more broadly. We will incorporate this into the revised manuscript.
>
> We hope these clarifications address your concerns, and we’re grateful for the opportunity to strengthen the paper further.

---

> ### Comment · Reviewer_RoyT · 2025-08-02
> **Reply to rebuttal**
>
> For Weaknesses 1 and 2:
> 1. I appreciate that the overall structure of the orientation maps remains robust despite changes in input strength. However, I am curious about how specific quantitative features, such as hypercolumn size and the nearest-neighbor distance between pinwheels, might vary with different levels of input. My opinion is a bit different from the answer that was given. Since input strength can influence neuronal firing rates, which directly lead to corresponding changes in these map metrics.
> 2. Thank you for reporting a pinwheel density of 3.63 and 3.46, as well as hypercolumn size and nearest-neighboring pinwheel distance. I have three brief follow-up points: #1. Your orientation maps look non-periodic and not fully converged—pinwheel density is only meaningful when hypercolumns tile periodically. Because some hypercolumns are large and some of them are small when this orientation map does not converge. I guess your pinwheel density value is only one observation of data. Could you clarify how you ensured convergence and whether a density measure is valid on your current maps? #2. The values you gave nearest-neighboring pinwheel distance lack units—are these in microns, millimetres, pixels, or something else? #3. Per Principles of Neural Science (Kandel et al.) and the Kaschube et al. (2010) / Ho et al. (2021) protocols I referenced, the pinwheel map has a very precise definition. If you mentioned the pinwheel map, it should have some metrics that meet the definition of the pinwheel map. Otherwise, you can only say that we have either the smooth-cluster maps or other types of maps. (This concern could be related to question 10)
>
> P.S. I suggest your new experimental data can be displayed in the form of tables, e.g., limitation 3.

---

> > ### Author Response · Authors · 2025-08-04
> >
> > Thank you for your response. Since both questions are closely related to the two metrics of the orientation map, i.e., the pinwheel density and the nearest-neighbour distance, we will address them together.
> >
> > This time, we revisited your comments carefully and realised that the metrics calculated in our case indeed may not be meaningful. As you noted, the orientation map in the Drosophila optic lobe is quite different from those in mammals. Evidences include that hypercolumns in Drosophila have quite different sizes and that pinwheel-like patterns are not evenly distributed in the map. We apologise for our oversight. Thus, we have decided to emphasise these differences between the orientation map found in Drosophila and those in mammals in the revised manuscript, and will not incorporate these metric values to avoid controversy.
> >
> > The term “orientation map” we use is a general concept in this field. It refers to a cell arrangement that contains pinwheel-like patterns, where the orientation preference of cells changes continuously around a singularity. Due to the discovery of such patterns in monkeys (Blasdel and Salama, 1986; Ts’o et al., 1990), cats (Bonhoeffer and Grinvald, 1991), tree shrews (Bosking et al., 1997), and others, the cell arrangements in the cortex of these animals were called “orientation maps” in the original works. As such, pinwheel-like patterns were also found in the Drosophila optic lobe, though not as evenly distributed as in mammals; we refer to this cell arrangement as an orientation map.
> >
> > The properties of the orientation map you mentioned, including the pinwheel density roughly equal to $pi$ across species (tree shew, ferret, primate) (Kaschube et al., 2010, Science; Ho et al., 2021, Current Biology) and nearest neighbour distance between pinwheels approximating 0.35L, are both very interesting. However, it is hard to say that they “define” the orientation map. They were obtained by analysing existing animal data, and it is unclear if they can be applied to all animals. In fact, we examined two textbooks, including one suggested by you (Kandel et al., Principles of Neural Science, 2013; Bear et al., Neuroscience: Exploring the Brain, 2007), but did not find that these properties are necessary for a cell arrangement to be called an “orientation map”.
> >
> > - Blasdel and Salama, Voltage-sensitive dyes reveal a modular organization in monkey striate cortex, Nature, 1986.
> > - Ts’o, et al., Functional organization of primate visual cortex revealed by high resolution optical imaging, Science, 1990.
> > - Bonhoeffer and Grinvald, Iso-orientation domains in cat visual cortex are arranged in pinwheel-like patterns, Nature, 1991.
> > - Bosking et al., Orientation selectivity and the arrangement of horizontal connections in tree shrew striate cortex, The Journal of Neuroscience, March 15, 1997.
> >
> > Nevertheless, thank you for your reminder. We will emphasise the large differences between the orientation map of Drosophila and those of other animals in the revised paper. We sincerely welcome any further valuable comments you may have.

---

> ### Comment · Reviewer_RoyT · 2025-08-05
>
> Thanks for providing some historical research about the orientation map. If the authors claim this orientation map is a "type of orientation map," which means there will be new clusters--not periodic and not fully converged-- in a fly's optic lobe. If this organization really exists, then my questions still have not been addressed. We know that a lot of higher mammals' visual cortex has periodic clustered pinwheels and pinwheel centers in the map, and how does an orientation map within pinwheel structures not follow a periodic pattern? Because the network of the visual cortex has the optimal E/I ratio, the network is balanced. Thus, the orientation representation is optimal, and the arrangement of this orientation map is periodic. If you said the new orientation map in a fly's optic lobe is not like this periodically, why? Can any new connectivity mechanisms be explained?
>
> Thus, I raised some questions but did not get a reply:
> >Input strength can influence neuronal firing rates, which directly lead to corresponding changes in these map metrics.
>
> So I just wondered if the input strength can influence neuronal firing rates, this firing rates will directly lead to corresponding map changes. The firing rates significantly influence the network, especially when the map is not periodic. In this context, some orientation preference neurons have a larger number, while others have a smaller number of neurons. When a larger number of these neurons are firing, they can compete with neurons that have different orientation preferences, leading to changes in the map arrangement. As the map changes, its arrangement will also be altered. Therefore, I conclude that this map from your work is not periodic and does not converge. If you continue to train this network, I guess the form of this map will change. If it does not change, can authors give me a reasonable answer?
>
> My second concern was that the pinwheel density value you gave is only one observation of data, because we need to statistically calculate all the pinwheels within their hypercolumn. So, the one observation of data is not convincing.
>
> My third question was whether the values you gave for the nearest-neighboring pinwheel distance lack units. And if the map is not periodic, how to get a fixed value? It should have very large and very small values.
>
> My fourth question was that I suggested your new experimental data in the rebuttal phase can be displayed in the form of tables to enhance the convincing, e.g., limitation 3.

---

> ### Author Response · Authors · 2025-08-06
>
> Thank you for your response.
>
> 1.	Frankly speaking, we are not very much sure if our understanding of the term “periodic” mentioned in your feedback is correct. In our understanding, “periodic” here refers to the even spatial distribution of pinwheels or regularly repeating patterns across the orientation map. Under this interpretation, our orientation map is indeed not periodic, and we do not claim that it should be. We don’t understand why a non-periodic orientation map is impossible. You mentioned that the visual cortex has an optimal E-I ratio, then the network is balanced, and the orientation map is periodic. We don’t understand why the optimal E/I ratio should lead to a periodic orientation map, specifically, the same number of neurons preferring different orientations (from your comment, this is a property of a periodic orientation map). Is this an established theory? In our understanding, E-I balance in a network does not necessarily indicate the same number of neurons preferring different orientations, considering that there are both excitatory and inhibitory neurons preferring similar orientations. Theoretically, it is possible that there is a dominant number of neurons preferring one orientation, and their E/I ratio is “optimal”. In addition, previous studies have reported biased orientation preferences in lower organisms like Drosophila [Straw et al. (2010); Zhao et al. (2025); Seelig et al. (2013)], which implies that those organisms may not use the same number of neurons to encode different orientations. Therefore, the lack of periodicity in our orientation map does not imply a violation of E-I balance, nor does it undermine the biological plausibility of our results.
>
> 2.	Regarding the input strength, our experiments showed that even when the stimulus statistics were altered (-20%, -40%), the orientation maps remained visually stable. This suggests that the maps are largely determined by the underlying circuit structure rather than being shaped dynamically by the input.
>
> 3.	You mention that our model “does not converge” and ask what will happen if we “continue to train this network”. This network was built based on the connectome of a static snapshot of a full adult fly (the network is actually copied from Shiu et al. paper (ref 45); the connectome from Zheng et al. (ref 49)), and the connection weights were fixed; there was NO training at all. So, whether the model “converges” does not apply here. This might be the main cause of your questions. We’re sorry we haven’t made this point clearer. If our understanding of “convergence” mentioned by you is incorrect, please correct us, and we will be happy to communicate with you further.
>
> 4.	Regarding your concern about “only one observation of data”, yes, as explained above, there is only one observation of data since only one adult fly connectome is available.
>
> 5.	Regarding the nearest-neighbour distance, as we mentioned in our previous response, we agree that this metric is not meaningful in our case. Since our orientation map is not periodic, there is no expectation of a consistent or characteristic spacing between pinwheels. As you have pointed out, such a map may contain both very large and very small inter-pinwheel distances, making it inappropriate to summarise with a single fixed value.
>
> 6.	Sorry, we missed this question. Here is a table summary of the ablation study:
>
>   number of [fit_quality=good] in ablation: X
>
>   number of [fit_quality=good] in original results: Y
>
>   X / Y = Z%
>
>   | Z      | silence Tm | silence Mi |
>   | ------ | ---------- | ---------- |
>   | **Dm** | 90.7%      | 2.4%       |
>   | **Pm** | 10.4%      | 3.9%       |
>
> Finally, it seems that you are very interested in the computational principles behind this special orientation map, e.g., how and why such a pattern emerges in a network, as the pattern is different from those observed in mammals, and existing theories seem not to be able to explain it. We agree that this is an interesting and important research topic for the future, but in this work, we focus on the discovery of this pattern.
>
> References:
>
> Straw, A. Visual control of altitude in flying Drosophila. Current Biology, 2010.
>
> Zhao, A. Eye structure shapes neuron function in Drosophila motion vision. Nature, 2025.
>
> Seelig, J.D. Feature detection and orientation tuning in the Drosophila central complex. Nature, 2013.

---

> ### Comment · Reviewer_RoyT · 2025-08-09
>
> Sorry for the delayed response. Periodicity of the orientation map is a fundamental concept. Orientation maps show a **repetitive layout** of iso-orientation domains interspersed with singularities. The arrangement of iso-orientation domains often approximates a hexagonal lattice locally, yet global variations and irregular pinwheel placements make the map **quasi-periodic** (Paik, 2011, Nature Neuroscience; Kaschube et al., 2010, Science). Another early theoretical work (Carreira-Perpiñán, 2005, Creb Cortex) reports cortical representations attempt to optimize a trade-off between coverage and continuity. Coverage versus continuity constraints minimize wiring cost to represent the visual information (Koulakov, 2001, Neuron; Swindale, 2000, Nature Neuroscience). Coverage ensures the orientation map **covers all orientation preferences** and preferred orientation changes continuously across the surface (continuity), forming iso-orientation domains where neighboring neurons share similar orientation tuning. However, in the authors' current work, only parts of the map satisfy both continuity and coverage; other parts exhibit continuity without coverage. This does not meet the requirements of quasi-periodicity, which means that in regions lacking coverage (only neurons surrounding pinwheel centers can cover all different orientations) can detect only some specific orientations and are **''blind'' to other orientations**. To this end, the fly's optic lobe would be **blind to some orientational features, which is counterintuitive**, and if the authors want to make the paper convincing, it needs stronger evidence to prove it.
>
> This is why I keep asking authors to clarify why their orientation map is not periodic and to report units for their metrics (e.g., nearest-neighboring pinwheel distance). If the nearest-neighboring pinwheel distance is very large (i.e., the path between pinwheels passes through extensive regions without pinwheel centers), or if the orientation map contains few pinwheel centers, the optic lobe may fail to ensure detection of every orientation.
>
> I hope this manuscript can make more convincing on this point, so I will keep my scores. By the way, I appreciate authors for their responses during this discussion phase.
>
>
> **References:**
>
> Paik, SB., Ringach, D. Retinal origin of orientation maps in visual cortex. Nat Neurosci 14, 919–925 (2011). https://doi.org/10.1038/nn.2824
>
> Kaschube M, Schnabel M, Löwel S, Coppola DM, White LE, Wolf F. Universality in the evolution of orientation columns in the visual cortex. Science. 2010 Nov 19;330(6007):1113-6. doi: 10.1126/science.1194869.
>
> Carreira-Perpiñán MA, Lister RJ, Goodhill GJ. A computational model for the development of multiple maps in primary visual cortex. Cereb Cortex. 2005 Aug;15(8):1222-33. doi: 10.1093/cercor/bhi004.
>
> Alexei A. Koulakov and Dmitri B. Chklovskii. Orientation Preference Patterns in Mammalian Visual Cortex: A Wire Length Minimization Approach. Neuron, 29(2):519–527, February 2001. doi: 10.1016/S0896-6273(01)00223-9
>
> Swindale, N., Shoham, D., Grinvald, A. et al. Visual cortex maps are optimized for uniform coverage. Nat Neurosci 3, 822–826 (2000). https://doi.org/10.1038/77731

---

### Official Review · Reviewer_mZ6e · 2025-07-01

**Clarity:** 3
**Significance:** 3
**Originality:** 3
**Rating:** 5
**Confidence:** 4

**Summary:**

This work computationally demonstrates the existence of orientation maps in the medulla of the drosophila visual system by simulating a network of leaky integrate and fire neurons with the measured connectome of drosophila. They convincingly showed the existence of such maps (at least in the computational model), and moreover showed that these maps have a shared columnar organization as a function of depth.  They even found a nice surprising result that early photoreceptors could show orientation selectivity and explained this in terms of recurrent loops from higher neurons.

**Questions:**

Please see my questions in the weaknesses section above.  I am giving this paper the benefit of the doubt, assuming the answer to these questions are satisfactory (especially the first two).  If not, I may have to lower my score.

A final question, which would be nice to discuss, are possible reasons for the diversity of the presence or absence of orientation maps across many species in the phylogenetic tree.  Is the presence or absence of smooth orientation maps, versus salt and pepper organization, a historical accident of evolution, or are there deep computational principles that dictate which will be found in which species?  How does the potential of smooth orientation maps in the fly impact the answer to this question?

**Ethical Concerns:**

["NO or VERY MINOR ethics concerns only"]

**Limitations:**

Yes. Subject to my questions in the weaknesses section above.

**Paper Formatting Concerns:**

None.

**Quality:**

3

**Strengths And Weaknesses:**

Strengths
1) The work is extremely thorough and careful.
2) It provides of a nice overview of orientation selectivity across multiple species suitable for a general reader.
3) It is well grounded in the biology of the drosophila nervous system.
4) It is an excellent example of going from a nonfunctional connectome to functional predictions.

Weaknesses
1) The robustness of the final results to parameter variations is left unexplored. For example, while many of the 10 parameters in appendix B.1 are set to fixed values found in the literature, it is unlikely that every neuron in the Drosophila brain has the same value for all parameters.  If there were an X% random change in all parameters (subject to sign constraints) how would the results degrade as a function of X?  If there were X% variations in each parameter from neuron/synapse to neuron/synapse, how would the results change?
2) It was unclear from reading Appendix B how w_{j,i} was chosen.  Is it simply 1 if there is a connection in the connectome, and 0 otherwise?  If so that should be made clear, as that would then be a strength of the paper.  If not, it should be explained how w_{j,i} was chosen and why - if this choice was sophisticated then more information than the connectome was used to define the simulation.  Also in equation 3 of appendix B should it be w_{syn} * w_{j,i} as the text suggests?
3) Finally, the existence of orientation selectivity/maps was already predicted qualitatively in prior cited work.  What does this quantitative measurement of the map in an insilico model add *conceptually* to our understanding above and beyond what has been said before?  A discussion of this point would be helpful.

---

> ### Author Rebuttal · Authors · 2025-07-31
>
> Thank you for the thoughtful and constructive feedback. Below, we address each of the specific concerns:
>
> 1.	Parameter robustness
>
> Good suggestion. Our model and parameters were taken from the LIF framework described in Shiu et al. (ref 45). To assess sensitivity to parameter perturbations, we performed a variation analysis where all parameters (except $V_{rest}$ and $V_{threshold}$) were randomly perturbed by a fixed percentage. At $\pm$10% variation, we observed minimal changes in preferred orientation across neurons. To further probe robustness, at $\pm$50% orientation selectivity was noticeably degraded; however, surprisingly, the ODEs (Eqns 1 and 2 in Appendix) remained stable and did not diverge. This suggests that the system retains a degree of functional robustness even under large perturbations. We will include results in the Appendix.
>
> 2.	Synaptic Weight Assignment
>
> Thank you for flagging this ambiguity. This computational method is taken from the Shiu et al. paper (ref 45). To clarify, $w_{j,i}$ represents the number of annotated synapses from neuron $j$ to neuron $i$ in the FAFB connectome. Thus, if a connection has multiple synapses, the conductance update is scaled proportionally.
>
> 3.	Equation 3
>
> Indeed, for equation 3 of Appendix B should be $w_{syn} * w_{j,i}$. We apologise for the lack of clarity and will revise the manuscript to clarify this formulation.
>
> 4.	Contribution Beyond Qualitative Predictions:
>
> While orientation selectivity has been qualitatively suggested in prior studies (e.g. Seung [ref 44]; Klapoetke et al. (2022)), these works either describe local orientation biases or propose conceptual models but do not demonstrate orientation maps. Our work differs in that we implement a spiking neural model derived entirely from the FAFB connectome without parameter tuning, and show that smooth orientation preference maps emerge naturally from the architecture. As detailed in lines 146-149, we further quantify the alignment between anatomical connectivity and simulated orientation preference, providing the first mechanistic and testable link between structure and function in this context.
>
> 5.	Cross-Species Perspective:
>
>  Our results suggest that orientation maps may not be exclusive to large-brained vertebrates, but arise from generic principles of connectivity and spatial organisation. The orientation maps in different species may have emerged independently across evolution, driven not by historical accident but by shared computational demands. E.g. trade-offs between wiring cost, spatial resolution, and the nature of downstream decoding. Further theoretical and computational studies are needed to reveal the underlying computational principles.
>
> We hope these clarifications address your concerns, and we’re grateful for the opportunity to strengthen the paper further.

---

> > ### Comment · Reviewer_mZ6e · 2025-08-01
> > **Reviewer response**
> >
> > Thank you - this response was helpful.  I recommend acceptance.

---

### Official Review · Reviewer_fCei · 2025-07-03

**Clarity:** 3
**Significance:** 3
**Originality:** 3
**Rating:** 5
**Confidence:** 2

**Summary:**

The paper presents a computaional model of the the brain of the fruit fly (Drosophila melanogaster). The authors simulate the responses of the neurons to different visual stimuli, and their analysis reveals the existence of orientation maps in the optic lobe, thus providing computational evidence for the previous hypothesis of such existence. This works shows that the orientation maps can emerge also in simpler non-mammal species visual systems.

**Questions:**

1. How did you chose 0.7 and 0.4 for the criteria of well-fit neurons? Can you provide a plot of R^2=0.7 (Similar to Figure 2-c)
2. How many total synapses and neurons did you model?

**Ethical Concerns:**

["NO or VERY MINOR ethics concerns only"]

**Final Justification:**

After reading the rebuttal I will remain with my original score and recommend acceptance of this paper.

**Limitations:**

yes.

**Paper Formatting Concerns:**

-

**Quality:**

3

**Strengths And Weaknesses:**

Strengths:
- the paper has novel scientific contribution by providing evidence of existence of orientation maps in the fruit fly vision system
- The authors provide detailed analysis of the results on neuron types level which can provide new insights to neuroscientists about the role of different cell types in visual perception.
- They use statistical measures and compare computational predictions with anatomical structure.

Weaknesses:
- As a NeurIPS audience, I am more interested in details of the computational modelling of the brain in the main paper, how many neurons and synapses were modelled in total, and novelties in this regard.
- The paper only tests orientation grating rather than natural scenes.
- The paper claims that the results offer insights to design of efficient scalable AI models (L 305). One should be cautious with such claims without any concrete suggestions or methods.
- Some related works are not discussed. See: "A functionally ordered visual feature map in the Drosophila brain", Klapoetke et.al.

Minor:
- Figure references are not linked to the figure. (Use \ref)

---

> ### Author Rebuttal · Authors · 2025-07-31
>
> Thank you for the constructive feedback. Below, we address the main concerns and clarify key points raised:
>
> 1.	We appreciate your interest in the modelling details. Our model includes a total of 138,639 neurons and 1,508,983 synapses (as in the connectome dataset), covering the full Drosophila brain connectome. We will revise the manuscript to state these details explicitly.
> 2.	Thank you for the suggestion of using Natural Stimuli rather than artificial stimuli. We agree that incorporating naturalistic stimuli may reveal other encoding mechanisms and is a valuable direction for future work. However, our current use of oriented bar stimuli was a deliberate design choice to isolate orientation tuning mechanisms.
> 3.	In response to the AI claim, thank you for the helpful suggestion. Our intent was to highlight the broader insight that minimal biological circuits can support structured visual representations. We will revise the relevant statement to reflect this more cautiously, emphasising biological significance over direct AI application.
> 4.	Regarding the missing relevant works, thank you for pointing this out. The study by Klapoetke et al. (2022) reveals functionally clustered representations of complex visual features in the lobular columnar neurons projecting to the central brain. In contrast, our work focuses on orientation selectivity in the earlier stages of the optic lobe, based on connectome-driven circuit structure. We will revise the manuscript to clarify how these complementary findings together highlight the presence of structured visual maps.
> Response to Specific Questions:
> 1.	The threshold was initially selected based on visual inspection during pilot analysis, aiming to balance fit quality and coverage. To justify this choice, we will include additional plots in the Appendix showing neurons across a range of R^2 and RSS_norm values, including examples near the threshold.
> 2.	As noted above, our connectome-driven model consists of a total of 138,639 neurons and 1,508,983 synapses. We will clarify these numbers more explicitly in the manuscript.
>
> We hope these clarifications address your concerns, and we’re grateful for the opportunity to strengthen the paper further.

---

> > ### Comment · Reviewer_fCei · 2025-08-04
> >
> > Thank you for the responses to my concerns. I look forward to seeing the plot of R^2=0.7 in the revision.

---

> ### Author Response · Authors · 2025-08-06
>
> Thank you for your thoughtful feedback. We'll make sure to include the updated plot in the revision.

---

### Note · Authors · 2025-08-12

We thank all reviewers for their constructive suggestions to further strengthen the paper. The following clarifications specifically address the points raised by Reviewer RoyT, and we thank you for your responses and feedback.

It is not the case that neurons in these regions fail to respond to other orientations. Rather, neurons whose strongest responses are to other orientations are fewer in number. At each spatial location, there is a population of neurons, and across this population many orientations are represented. However, in our analysis, each location is assigned only the orientation corresponding to the strongest response in that population.

We also wish the emphasise that the focus of this work is to report the presence of such an orientation map in the fly visual system. The understanding of the underlying mechanisms, and why the pattern differs from that of mammals, are indeed fascinating questions, but they fall outside the scope of the present study.

---

### Decision · Program_Chairs · 2025-09-17

**Decision:**

Accept (poster)

**Comment:**

This paper simulates orientation maps in Drosophila optic lobe using the full measured brain connectome of Drosophila and networks of simulated spiking neurons. In a computational modeling effort, they address an interesting open problem in neuroscience: whether topographic orientation maps, which have been mainly observed in mammalian visual systems, require cortical structures or can also emerge in species without cortical structure. The reviewers recognized several significant strengths in this work, including very thorough experiments and analysis, and the compelling approach of connecting connectome data to functional observations. The modeling work provides valuable insights suggesting that topographic orientation maps do not necessarily require cortical structures and can emerge across species with large phylogenetic distances.

The reviewers initially raised a few concerns such as the sensitivity of results to parameters and ambiguities in parameter selection. However, these issues appear to have been adequately addressed in the authors' rebuttals. One reviewer also raised concerns regarding differences between the structure of simulated orientation maps in Drosophila compared to those observed in mammalian visual systems. While the authors acknowledged these differences, they should address them more clearly in the camera-ready version. Nevertheless, I believe these cross-species differences could provide interesting experimental predictions for future comparative studies.

Overall, this work makes a solid contribution to computational neuroscience by addressing a fundamental question about visual system organization across species. The thorough experimental approach and successful bridging of structural and functional data outweigh the remaining concerns. Therefore, I recommend acceptance.